# Perturbations in eIF3 subunit stoichiometry alter expression of ribosomal proteins and key components of the MAPK signaling pathways

Anna Herrmannová[1]*[†], Jan Jelínek[2][†], Klára Pospíšilová[1], Farkas Kerényi[1], Tomáš Vomastek[3], Kathleen Watt[4], Jan Brábek[5], Mahabub Pasha Mohammad[1], Susan Wagner[1], Ivan Topisirovic[6], Leoš Shivaya Valášek[1]*

[1]Laboratory of Regulation of Gene Expression, Institute of Microbiology of the Czech Academy of Sciences, Prague, Czech Republic; [2]Laboratory of Bioinformatics, Institute of Microbiology of the Czech Academy of Sciences, Prague, Czech Republic; [3]Laboratory of Cell Signaling, Institute of Microbiology of the Czech Academy of Sciences, Prague, Czech Republic; [4]Science for Life Laboratory, Department of Oncology-Pathology, Karolinska Institutet, Solna, Sweden; [5]Lady Davis Institute, Laboratory of Cancer Cell Invasion, Faculty of Science, Charles University, Prague, Czech Republic; [6]Lady Davis Institute, Gerald Bronfman Department of Oncology, Department of Biochemistry, Division of Experimental Medicine, McGill University, Montréal, Canada

*For correspondence:
herrmannova@seznam.cz (AH);
valasekl@biomed.cas.cz (LSV)

[#] These authors contributed equally to this work.

## eLife Assessment

This study demonstrates mRNA-specific regulation of translation by subunits of the eukaryotic initiation factor complex 3 (eIF3) using **convincing** methods, data, and analyses. The investigations have generated **important** information that will be of interest to biologists studying translation regulation. However, the physiological significance of the gene expression changes that were observed is not clear.

**Abstract** Protein synthesis plays a major role in homeostasis and when dysregulated leads to various pathologies including cancer. To this end, imbalanced expression of eukaryotic translation initiation factors (eIFs) is not only a consequence but also a driver of neoplastic growth. eIF3 is the largest, multi-subunit translation initiation complex with a modular assembly, where aberrant expression of one subunit generates only partially functional subcomplexes. To comprehensively study the effects of eIF3 remodeling, we contrasted the impact of eIF3d, eIF3e or eIF3h depletion on the translatome of HeLa cells using Ribo-seq. Depletion of eIF3d or eIF3e, but not eIF3h reduced the levels of multiple components of the MAPK signaling pathways. Surprisingly, however, depletion of all three eIF3 subunits increased MAPK/ERK pathway activity. Depletion of eIF3e and partially eIF3d also increased translation of TOP mRNAs that encode mainly ribosomal proteins and other components of the translational machinery. Moreover, alterations in eIF3 subunit stoichiometry were often associated with changes in translation of mRNAs containing short uORFs, as in the case of the proto-oncogene MDM2 and the transcription factor ATF4. Collectively, perturbations in eIF3 subunit stoichiometry exert specific effect on the translatome comprising signaling and stress-related transcripts with complex 5′ UTRs that are implicated in homeostatic adaptation to stress and cancer.

## Introduction

Translational control contributes significantly to the overall regulation of gene expression. Indeed, regulation at the level of translation provides quicker adaptive and usually reversible responses to environmental cues than transcriptional regulation as it acts on pre-existing mRNAs by simply turning their expression on or off (*Sonenberg and Hinnebusch, 2009*). As such, translational control critically contributes to the maintenance of overall cellular homeostasis, suggesting that its dysregulation has detrimental effects on cell proliferation and, more broadly, on health (*Scheper et al., 2007*). In a rapidly growing number of cases, mutations and/or impaired expression of many translational components have been shown to either directly lead to or significantly contribute to various pathological conditions, including cancer (reviewed in *de la Parra et al., 2018b*; *Robichaud and Sonenberg, 2017*; *Spilka et al., 2013*; *Sriram et al., 2018*).

Translation can be divided into four stages with the initiation stage playing the most critical role with respect to translational control. It begins with the formation of the 43 S pre-initiation complex (PIC) composed of the 40 S ribosomal subunit, the eIF2*GTP*Met-tRNA$_i^{Met}$ ternary complex (eIF2-TC), and eIFs 1, 1 A, 3, and 5 (reviewed in *Valásek, 2012*). The next step is the mRNA recruitment to form the 48 S PIC, which is facilitated by the eIF4F cap-binding complex in concert with eIF3. Once the mRNA is loaded, the 48 S PIC starts scanning the mRNA's 5′ untranslated region (UTR) until the start codon has been recognized by base-pairing with the Met-tRNA$_i^{Met}$ anticodon. This drives conformational changes in the 48 S PIC, co-operatively mediated by eIF1, eIF1A, eIF2, eIF3 and eIF5, resulting in the closure of the 40 S mRNA binding channel and subsequent ejection of the eIF2*GDP*eIF5 subcomplex to allow subunit joining to commence (reviewed in *Hinnebusch, 2017*).

Cellular stresses primarily inhibit cap-dependent translation initiation *via* the eIF4F and eIF2-TC complexes. The former is under the control of two major stress-signaling hubs – the anabolic kinase – the mechanistic target of rapamycin complex 1 (mTORC1) and the mitogen-activated protein kinases (MAPKs). They promote growth under nutrient-rich and stress-free conditions and deactivate in response to limited supply of amino acids, oxygen, ATP, growth factors or nutrients and play a role in the unfolded protein response (*Bhat et al., 2015*; *Lindqvist et al., 2018*; *Roux and Topisirovic, 2018*). These pathways are hyperactive in the vast majority of cancers, mainly due to activating mutations of RAS, BRAF and PI3K or loss-of-function mutations of the tumor suppressor PTEN, which are among the most common genetic perturbations in cancer (*Kandoth et al., 2013*). The latter – eIF2-TC – is a major target of the pro-survival, adaptive pathway called integrated stress response (ISR) encompassing four members of the eIF2α kinase family (PERK, PKR, GCN2, and HRI) that respond to a multitude of stimuli including nutrient limitation and endoplasmic reticulum stress by rapidly shutting down general translation (*Pakos-Zebrucka et al., 2016*).

Mitogen-activated protein kinase (MAPK) cascades have been shown to play a key role in the transduction of extracellular signals into cellular responses. Three families of MAPK signaling pathways have been well characterized in mammalian cells: extracellular signal-regulated kinase (ERK), c-Jun N-terminal kinase/stress-activated protein kinase (JNK/SAPK), and p38 kinase (*Cargnello and Roux, 2011*; *Zhang and Liu, 2002*). The activation of MAPKs is a multistep process. In case of the best described Raf-MEK-ERK pathway, the essential linkers of epidermal growth factor receptors and other receptor tyrosine kinases with the MAPK cascade include adaptor protein Grb2, a Son of Sevenless (SOS) family of guanine nucleotide exchange factors, and a small GTP binding protein Ras (*Figure 1A*). The MAPK cascade begins with MAPKKK (represented by c-Raf) which phosphorylates and activates MEK1 and MEK2 (MAPKKs). MEKs eventually phosphorylate, and thereby activate p44 MAPK and p42 MAPK, also known as ERK1 and ERK2, respectively (*Lavoie et al., 2020*; *Stokoe et al., 1994*). The activated ERKs then translocate to the nucleus where they phosphorylate and transactivate various transcription factors such as Elk-1, c-Jun, and c-Fos, thereby altering gene expression to promote growth, differentiation, or mitosis. The ERK/MAPK pathway also directly regulates translational machinery, for example by stimulating the activity of both ribosomal S6 protein kinases (RSKs; RSK1-4 in humans; *Carriere et al., 2008*) and MAPK-interacting kinases (MNKs; MNK1 and 2 in humans; *Buxade et al., 2008*; *Fukunaga and Hunter, 1997*; *Waskiewicz et al., 1997*). MNKs interact with eIF4G and phosphorylate eIF4E at Ser209, a site that increases the oncogenic potential of eIF4E (*Topisirovic et al., 2004*) and facilitates translation of specific mRNAs related to migration, metastasis and inflammation (*Furic et al., 2010*; *Pyronnet et al., 1999*; *Robichaud et al., 2015*). Following the stimulation of the ERK pathway, activated RSKs phosphorylate rpS6, eIF4B, eEF2K, and PDCD4, which are all important

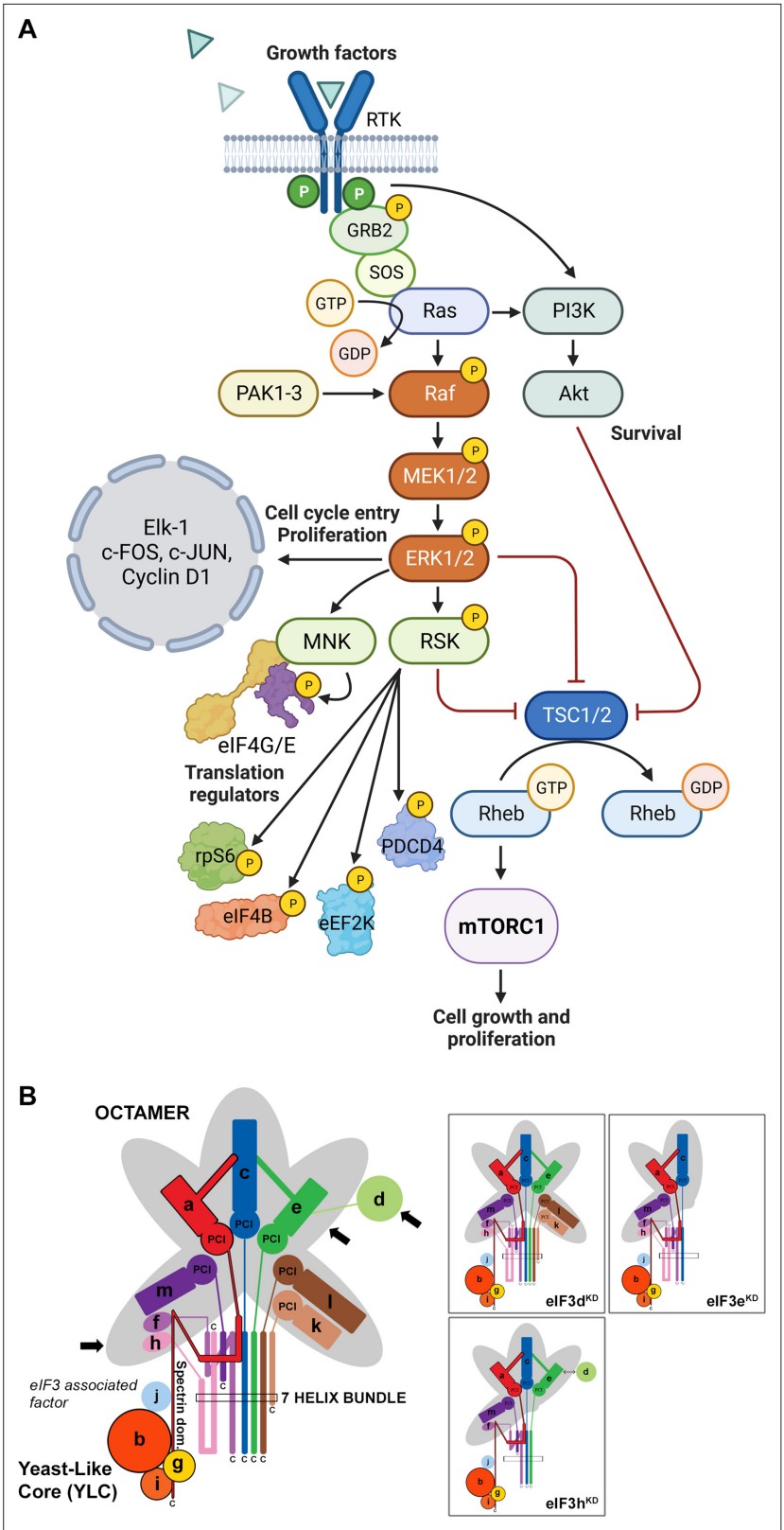

**Figure 1.** Schematic models of the core of MAPK/ERK pathway and human eIF3 complex. (**A**) The MAPK/ERK pathway activation by growth factors and receptor tyrosine kinases (RTKs). RTKs signal through GRB2-Sos and Ras to activate Raf-MEK-ERK signaling cascade. Active ERK phosphorylates numerous substrates, both nuclear and cytoplasmic, to affect gene transcription, protein translation, cell growth, proliferation, and survival. Both RTKs and

*Figure 1 continued on next page*

*Figure 1 continued*

Ras also activate PI3K-AKT pathway that affect cell survival and growth. Conventional PAKs (PAK1-3) phosphorylate c-Raf and contribute to its activation. (**B**) eIF3 subunits forming the PCI/MPN octamer are indicated by the grey background. The rectangle marks the seven α-helices involved in formation of the 7-helix bundle. The Yeast-Like Core (YLC) comprising the eIF3 subunits a, b, g, and i is depicted, and so is the eIF3-associated factor eIF3j. Arrows indicate subunits targeted by siRNA and subjected to Ribo-Seq in this study; eIF3 subcomplexes generated in individual knock-downs are boxed (adapted from Figure 6 of *Wagner et al., 2016*). This figure was created with BioRender.com.

The online version of this article includes the following source data and figure supplement(s) for figure 1:

**Figure supplement 1.** Quality control of eIF3e, d, and h knock-downs - Hela cells downregulated for eIF3d, eIF3e, and eIF3h display slower growth and impaired translation.

**Figure supplement 1—source data 1.** Original files for western blot analysis displayed in *Figure 1—figure supplement 1C*.

**Figure supplement 1—source data 2.** PDF file containing original western blots for *Figure 1—figure supplement 1C*, indicating the relevant bands and treatments.

translational regulators (reviewed in *Bhat et al., 2015*; *Robichaud and Sonenberg, 2017*). RSKs have also been implicated in the regulation of other signaling pathways involved in growth control, stress-response and cancer including the regulation of mTORC1 (reviewed in *Roux and Topisirovic, 2018*). For example, RSKs have been shown phosphorylate and inhibit tuberous sclerosis complex 2 (TSC2) that acts as the negative regulator of mTORC1 signaling (*Roux et al., 2004*, *Figure 1A*).

It has been shown that alterations in genomic sequences, protein levels, stoichiometry, and/or activity of eIFs and other components of the translational apparatus and relevant signaling cascades, as well as in cis-regulatory elements of mRNAs, can circumvent the stress-induced translational shut-down that these pathways impose by allowing selective recruitment of mRNAs, rewiring translational control of mRNA cis-regulatory elements, and/or enabling alternative start codon selection (*Kovalski et al., 2022*; *Robichaud et al., 2019*; *Sriram et al., 2018*). Naturally, this leads to various pathologies. Overexpression of eIF4F complex components, for example, has been observed in various cancer entities (*Bhat et al., 2015*; *De Benedetti and Harris, 1999*; *Lazaris-Karatzas et al., 1990*; *Ruggero, 2013*; *Shuda et al., 2000*; *Truitt and Ruggero, 2016*). In addition to eIF4F components, dysregulation of the multi-subunit eIF3 complex is also involved in oncogenesis. Ten of the 12 eIF3 subunits have been linked to human cancers, many due to overexpression (a, b, c, d, g, h, m or i), some due to underexpression (e and f) (reviewed in *de la Parra et al., 2018b*; *Gomes-Duarte et al., 2018*; *Hershey, 2015*; *Valášek et al., 2017*; *Zhang et al., 2007*). In this regard, it is worth stressing that some eIF3 subunits have been shown to directly interact with specific mRNAs (*Lee et al., 2015*), eIF3d together with DAP5 was shown to facilitate alternate form of cap-dependent mRNA translation (*de la Parra et al., 2018a*), and eIF3k was recently proposed to serve as a regulator of the balanced ribosome content (*Duan et al., 2023*). Despite all these observations, the exact mechanism(s) by which eIF3 affects various pathologies is virtually unknown. Moreover, previous studies that investigated the effect of eIF3 in specific diseases focus on altered levels of only one of the eIF3 subunits, while neglecting the need to consider the context of altered expression of the entire eIF3 complex.

Here, we draw attention to this unfortunate trend. eIF3 is the largest and most complex eIF of all eIFs, it is modular in nature (*Figure 1B*) (reviewed in *Valášek et al., 2017*; *Wolf et al., 2020*), and plays multiple roles not only in translation initiation, but also in termination, ribosomal recycling and stop codon readthrough (*Beznosková et al., 2013*; *Beznosková et al., 2015*; *Pisarev et al., 2007*; *Poncová et al., 2019*). Each of these roles may be regulated differently by individual subunits and/or their combinations, making it practically impossible to study them in isolation. Eight eIF3 subunits (a, c, e, f, h, k, l, and m) form a structural scaffold called the PCI (**P**roteasome, **C**OP9, e**I**F3) / MPN (**M**pr1-**P**ad1 **N**-terminal) octamer, whereas the remaining four non-octameric subunits (b, d, g, and i) are likely to be more flexible. The essential subunits b, g, and i form a separate module that is connected to the octamer *via* the C-terminal domain (CTD) of eIF3a; together they are called YLC for **Y**east **L**ike **C**ore (*Wagner et al., 2016*). The eIF3d subunit is located on the eIF3 periphery and is attached to the octamer primarily via eIF3e, but partly also *via* eIF3a and c (*Bochler et al., 2020*; *Brito Querido et al., 2020*). On the 40 S subunit, the octamer is positioned close to the mRNA exit site, whereas the mobile

YLC shuffles between the mRNA entry site and the subunit interface during the AUG selection process (*des Georges et al., 2015*; *Llácer et al., 2021*; *Simonetti et al., 2020*).

We recently reported a comprehensive in vivo analysis of the modular dynamics of the human eIF3 complex (*Wagner et al., 2020*; *Wagner et al., 2014*; *Wagner et al., 2016*). Using a systematic individual downregulation strategy, we showed that the expression of all 12 eIF3 subunits is interconnected such that perturbance of the expression of one subunit results in the down-regulation of entire modules leading to the formation of partial eIF3 subcomplexes with limited functionality (*Herrmannová et al., 2020*). eIF3d is the only exception in this respect, as its downregulation does not influence expression of any other eIF3 subunit. Here, we took advantage of this knowledge to examine the specific effect of selected eIF3 subcomplexes on the translation efficiency, transcriptome-wide, in HeLa cells.

## Results
### High-quality sequencing libraries prepared from HeLa cells expressing different eIF3 subcomplexes

To investigate the impact of different human eIF3 subunits and partial eIF3 sub-complexes, which arise in cells as a result of unbalanced eIF3 expression, on translational efficiency transcriptome-wide, we performed Ribo-Seq together with RNA-Seq, which allowed us to draw conclusions about gene expression at the transcriptional and translational levels and their interdependence. In particular, using the siRNA technology, we examined HeLa cells with reduced levels of eIF3d, eIF3e, or eIF3h subunits and cells treated with non-targeting (NT) siRNA. These eIF3 subunits were selected because the eIF3d depletion leads to a reduction in protein levels of only eIF3d and leaves the rest of the eIF3 complex intact, whereas downregulation of eIF3e or eIF3h generates partial eIF3 subcomplexes as previously described (*Wagner et al., 2016*). Specifically, targeting eIF3e with siRNA results in reduced protein levels of eIF3e, eIF3d, eIF3k, and eIF3l, whereas downregulation of eIF3h produces a partial eIF3 complex lacking subunits eIF3h, eIF3k, and eIF3l (*Figure 1B*). In terms of impact on cell growth, both eIF3d$^{KD}$ and eIF3e$^{KD}$ strongly reduce proliferation, while the effect of eIF3h$^{KD}$ is only moderate (*Figure 1—figure supplement 1A*, *Wagner et al., 2016*).

We first performed several control experiments with aliquots of cell lysates made to prepare individual Ribo-Seq libraries to document an efficient downregulation of the targeted eIF3 subunit, as well as the expected phenotype (*Wagner et al., 2016*). Using qPCR, we measured mRNA levels of the siRNA targeted eIF3 subunits (eIF3d, eIF3e, eIF3h), as well as of eIF3b as a negative control and a housekeeping gene ALAS1 for normalization purposes. Routinely, a difference of 4–5 cycles was achieved for each downregulated subunit, corresponding to ~16–32 fold decrease in mRNA levels; the mRNA levels of the control subunit eIF3b remained unchanged (*Figure 1—figure supplement 1B*).

Next, we determined protein levels of selected eIF3 subunits by Western blotting. Protein levels of all eIF3 subunits targeted individually by siRNA were strongly reduced (*Figure 1—figure supplement 1C*). Depletion of eIF3e also induced a marked decrease in the eIF3d, eIF3k, and eIF3l protein levels (*Figure 1—figure supplement 1C* and data not shown), as previously reported (*Wagner et al., 2016*). Finally, we performed polysome profiling to assess the impact on translation and obtained essentially the same results as before (*Wagner et al., 2016*). The eIF3d$^{KD}$ showed the most pronounced defect in translation initiation, marked by a robust increase of the 80 S monosome peak and a concomitant decrease of polysome content, similar to eIF3e$^{KD}$, while eIF3h$^{KD}$ had only a modest effect on polysome formation (*Figure 1—figure supplement 1D*).

Ribo-Seq libraries were generated from each knock-down and NT control in three biological replicates as described in *Ingolia et al., 2012*. RNA-Seq libraries were prepared in four biological replicates using the SMARTer smRNA-Seq Kit to normalize for mRNA levels and calculate translational efficiency (TE). Both Ribo-Seq and RNA-seq libraries were highly reproducible as shown by the Spearman correlation coefficient (*Figure 2—figure supplement 1A*). All Ribo-Seq libraries showed a similar read length distribution with a prominent peak at 30–32 nt and an expected 3 nt periodicity in coding sequences (CDS; *Figure 2—figure supplement 1B, C* and *Figure 2—figure supplement 2A*). As expected, Ribo-seq libraries were strongly enriched for CDS reads compared to RNA-Seq libraries (*Figure 2—figure supplement 2B, C*). The PCA plot and hierarchical clustering (*Figure 2A* and

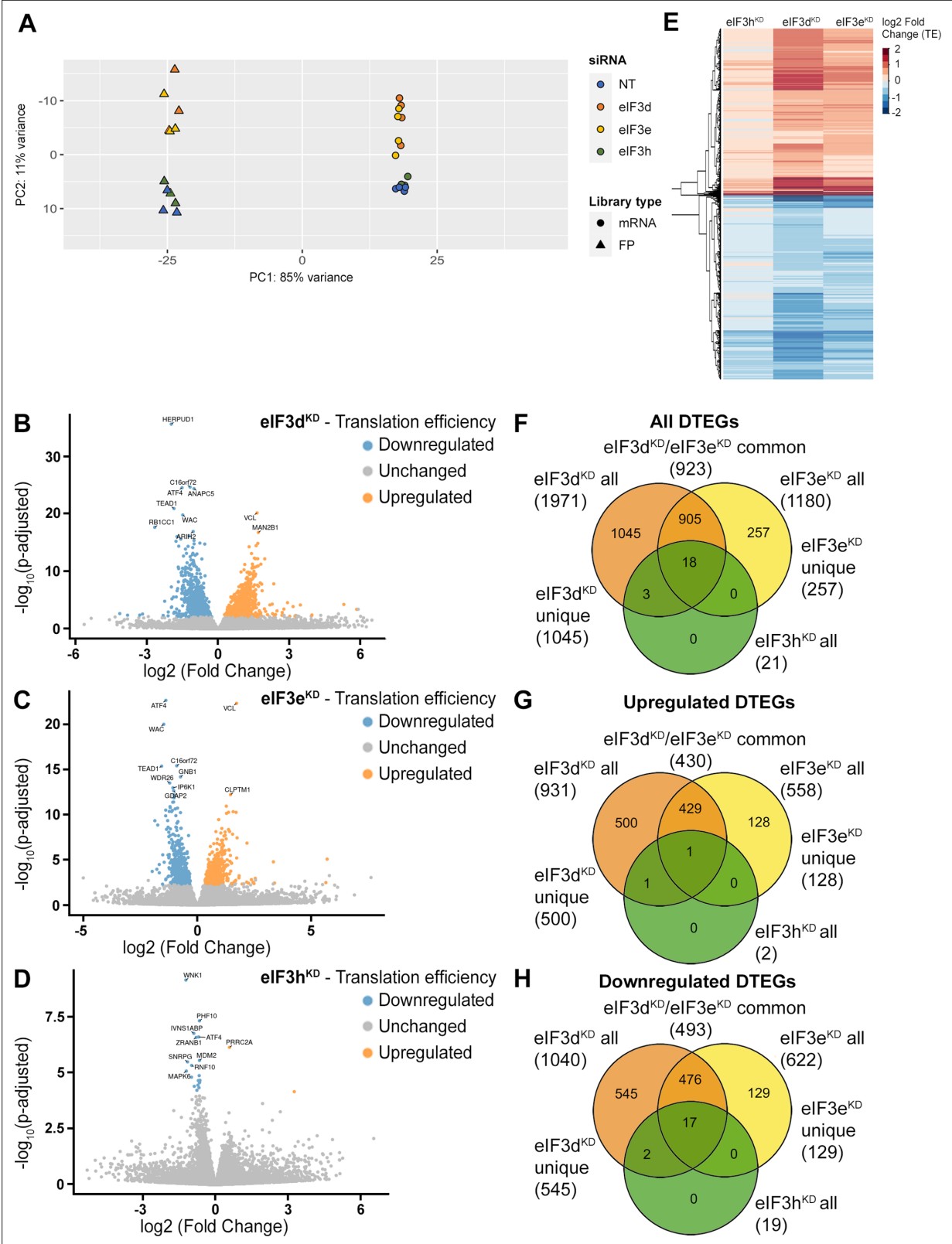

**Figure 2.** DTEGs identified in eIF3d$^{KD}$, eIF3e$^{KD}$, and eIF3h$^{KD}$ largely overlap. (**A**) Principal component analysis of read count per gene of the RNA-Seq and Ribo-Seq libraries. (**B** - **D**) Volcano plots of significant ($P_{adjusted}$ <0.05) DTEGs in individual knock-downs: eIF3d$^{KD}$ (**B**), eIF3e$^{KD}$ (**C**) and eIF3h$^{KD}$ (**D**). The volcano plots show the Log2 fold-change of TE (x-axis) *versus* the significance -Log10 p-adjusted value (y-axis). Genes without significant p-adjusted value are plotted in grey, downregulated DTEGs are plotted in blue and upregulated DTEGs in orange. For each plot, the top 10 most significant

*Figure 2 continued on next page*

*Figure 2 continued*

DTEGs are labeled with gene names. The Volcano plots were generated using modified script from Galaxy (*Blankenberg et al., 2014*). (**E**) Heatmap and dendrogram resulting from hierarchical clustering analysis of significant TE changes observed in eIF3d$^{KD}$, eIF3e$^{KD}$, and eIF3h$^{KD}$ cells. (F - H) Venn diagrams depicting the overlap in all DTEGs (**F**), in upregulated DTEGs (**G**), and in downregulated DTEGs (**H**) among eIF3d$^{KD}$, eIF3e$^{KD}$, and eIF3h$^{KD}$. Based on the overlaps, DTEGs were divided in several subgroups for further analysis: eIF3d$^{KD}$ 'all', eIF3d$^{KD}$ 'unique', eIF3e$^{KD}$ 'all', eIF3e$^{KD}$ 'unique', eIF3d$^{KD}$/eIF3e$^{KD}$ 'common' and eIF3h$^{KD}$ 'all'.

The online version of this article includes the following figure supplement(s) for figure 2:

**Figure supplement 1.** Quality control of Ribo-Seq libraries from all eIF3 knock-downs and the NT control.

**Figure supplement 2.** Ribo-Seq libraries from all eIF3 knock-downs and the NT control display triplet periodicity and enrichment of reads in CDS.

**Figure supplement 3.** Clustering of Ribo-Seq and RNA-Seq libraries.

**Figure supplement 4.** DTEGs identified in all knock-downs largely overlap.

*Figure 2—figure supplement 3*) showed clustering of the samples into two main groups: Ribo-Seq and RNA-seq, and also into two subgroups; NT and eIF3h$^{KD}$ samples clustered on one side and eIF3e$^{KD}$ and eIF3d$^{KD}$ samples on the other. These results suggest that the eIF3h depletion has a much milder impact on the translatome than depletion of eIF3e or eIF3d, which agrees with the growth phenotype and polysome profile analyses (*Figure 1—figure supplement 1A, D*).

## eIF3 subcomplexes alter translation efficiency of a large proportion of genes

Using the R DESeq2 package (*Love et al., 2014*), we identified transcripts with significantly altered TE ($P_{adjusted}$ <0.05) for each knock-down compared to the NT control (*Figure 2B, D*, *Supplementary file 1*), henceforth referred to as Differential Translation Efficiency Genes (DTEGs) (*Chothani et al., 2019*). In total, we identified 21 DTEGs in eIF3h$^{KD}$ (19 downregulated and 2 upregulated), 1180 DTEGs in eIF3e$^{KD}$ (622 down and 558 up), and 1971 DTEGs in eIF3d$^{KD}$ (1040 down and 931 up) out of a total of 17520 genes that were assigned TE values.

The number of DTEGs in each knock-down correlated well with the severity of the overall phenotype and with the impact on global translation, as judged from the polysome profiles. eIF3d$^{KD}$ with the strongest decrease in polysomes had the highest number of DTEGs, whereas eIF3h$^{KD}$ with only 21 DTEGs had a marginal effect on polysome formation (*Figure 1—figure supplement 1D*). Consistently, the overall distribution of fold-changes in TE showed most pronounced changes in eIF3d$^{KD}$ (*Figure 2E*). It should be noted, however, that in contrast to polysome profiling, differential expression analysis of Ribo-Seq data normalizes out global changes in translational rates, and reveals only specific shifts of ribosome localization on individual mRNAs between the NT and KD datasets. Therefore, the fact that all three knock-downs showed an increase in TE (not just a decrease) for approximately half of the affected genes, clearly points to an impairment of a specific mode of regulation rather than simply to a non-specific global translational shutdown.

Interestingly, there is a considerable overlap between the knock-downs studied, such that only 1045 DTEGs are unique for eIF3d$^{KD}$ and the rest (926) are shared with other two knock-downs; in particular 923 with eIF3e$^{KD}$ and 21 with eIF3h$^{KD}$ – in the latter case, the overlap reaches 100% (*Figure 2F, Figure 2—figure supplement 4A–C*). Remarkably, eIF3d$^{KD}$ affects neither the expression of any other eIF3 subunit nor the integrity of the eIF3 complex, yet it has the most pronounced impact on translation (*Wagner et al., 2016*), correlating well with the highest number of DTEGs identified. In case of eIF3e$^{KD}$, only 257 DTEGs are unique and the rest (923) are shared with eIF3d$^{KD}$, 18 of which are also shared with eIF3h$^{KD}$ (*Figure 2F, Figure 2—figure supplement 4A–C*). The eIF3 subcomplex formed in eIF3e$^{KD}$ lacks subunits e, k and l and partially also eIF3d (*Wagner et al., 2016*) – the significantly reduced protein levels of eIF3d are evident in *Figure 1—figure supplement 1C*. Thus, it is not surprising that the DTEGs identified in eIF3e$^{KD}$ largely overlap with those identified in eIF3d$^{KD}$; in fact, this result supports the robustness and accuracy of this approach. The fact that the magnitude of the knock-down effect is bigger in eIF3d$^{KD}$ over eIF3e$^{KD}$ is discussed below.

## eIF3d^KD and eIF3e^KD increase TE of mRNAs encoding proteins associated with membrane organelles

To analyze enrichment in KEGG pathways and GO Biological Processes in a meaningful way, we analyzed the DTEGs divided not only into upregulated and downregulated groups, but also into specific subgroups – eIF3d^KD 'all', eIF3e^KD 'all', eIF3d^KD 'unique', eIF3e^KD 'unique' and eIF3d^KD/eIF3e^KD 'common' (*Figure 2F–H*). To disentangle the effects of the changes in mRNA abundance vs. translational regulation, we also investigated translationally exclusive DTEGs whose mRNA abundance did not change significantly and therefore the change in TE can be attributed solely to translational regulation; eIF3d^KD 'translation only' and eIF3e^KD 'translation only', as well as translationally buffered DTEGs whose mRNA abundance changes in opposite direction than translational regulation, that is mRNA increases but TE decreases so the net protein production is not changed or even decreased; eIF3d^KD 'buffered' and eIF3e^KD 'buffered' (*Figure 2—figure supplement 4D–F*). For schematic illustrating classification of genes based on fold changes of footprints (FP), mRNA, and TE, see *Figure 2—figure supplement 4G, H*.

Analysis of DTEGs upregulated in eIF3d^KD and eIF3e^KD showed similar results for almost all groups analyzed (eIF3d^KD 'all', 'unique', and 'translation only', eIF3d^KD/eIF3e^KD 'common', as well as eIF3e^KD 'all' and 'translation only'; *Figure 3A, B*, and *Figure 3—figure supplement 1A*), with the strongest enrichment in the KEGG 'Lysosome' and 'Protein processing in endoplasmic reticulum' pathways. The majority of DTEGs in these pathways were solely translationally regulated meaning that their mRNA levels stayed unchanged while the TE increased. For example, for the KEGG pathway 'Lysosome', more than 70% (i.e. 37 out of the 51) DTEGs upregulated in the eIF3d^KD 'all', fall into the eIF3d^KD 'translation only' group. This suggests that when translation initiation is compromised by eIF3d/e^KD, cells may increase protein degradation and turnover by translationally upregulating lysosomal and ER proteins. Interestingly and in accord with our results, Lin et al. detected increased lysosomal load in MCF-10A cells downregulated for eIF3e (*Lin et al., 2020*). They speculated that this is due to the need to clear defective mitochondria. In support, we also found enrichment of mitochondria-associated DTEGs, especially mitochondrial ribosomal proteins (see below).

Among the eIF3d^KD 'unique upregulated' DTEGs, we identified one interesting KEGG pathway, the ABC transporters, which did not show up in other gene groups (*Figure 3—figure supplement 1A*, in green). A total of 12 different ABC transporters had elevated TE (9 of them are unique to eIF3d^KD, while 3 were also found in eIF3e^KD), 6 of which (ABCC1-5, ABCC10) belong to the C subfamily, known to confer multidrug resistance with alternative designation as multidrug resistance protein (MRP1-5, MRP7; *Sodani et al., 2012*). Interestingly, all six of these ABCC transporters were upregulated solely at the translational level (*Supplementary file 1*).

## eIF3e^KD increases production of ribosomal proteins

Interestingly, the KEGG pathway 'Ribosome' and GO terms connected with translation and ribosome biogenesis were enriched amongst DTEGs upregulated specifically in eIF3e^KD 'unique upregulated' DTEGs (*Figure 3C* and *Figure 3—figure supplement 1B*). We then compared the significance of enrichment for 'Ribosome' KEGG term between eIF3e^KD and eIF3d^KD 'all upregulated' groups. This was the only case where a term or a pathway was more significantly enriched in eIF3e^KD (p=5.199E-10) than in eIF3d^KD (NS, p=0.1183) (*Figure 3—figure supplement 1A*, in green). Intrigued by this result, we filtered out all ribosomal proteins (RPs) from our data set (according to HGNC database, *Tweedie et al., 2021*), regardless of the TE changes, which consisted of 89 cytoplasmic and 77 mitochondrial RPs, and analyzed them separately (*Supplementary file 1*). In total, we identified 32 RPs with significantly changed TE in eIF3e^KD (14 cytoplasmic and 18 mitochondrial) and only 16 in eIF3d^KD (9 cytoplasmic and 7 mitochondrial). Moreover, all the cytoplasmic RPs with increased TE in eIF3e^KD (and most mitochondrial ones) were upregulated solely at the translational level and their mRNA levels were not changed. We selected several RPs with different combinations of changes in TE or FP or mRNA levels in both eIF3e^KD and eIF3d^KD (*Figure 3D*) and subjected them to western blotting. All ribosomal proteins tested showed visibly increased protein levels in both eIF3e^KD and eIF3d^KD (*Figure 3E, F*), while their mRNA levels remained unchanged (*Figure 3—figure supplement 1C*).

As the production of the ribosomes seemed to be increased, we next examined the 60 S/40 S subunit ratio to control for unbalanced ribosome biogenesis. We performed polysome profiling in the presence of EDTA that splits all ribosomes into 40 S and 60 S subunits and showed that the 60 S/40 S

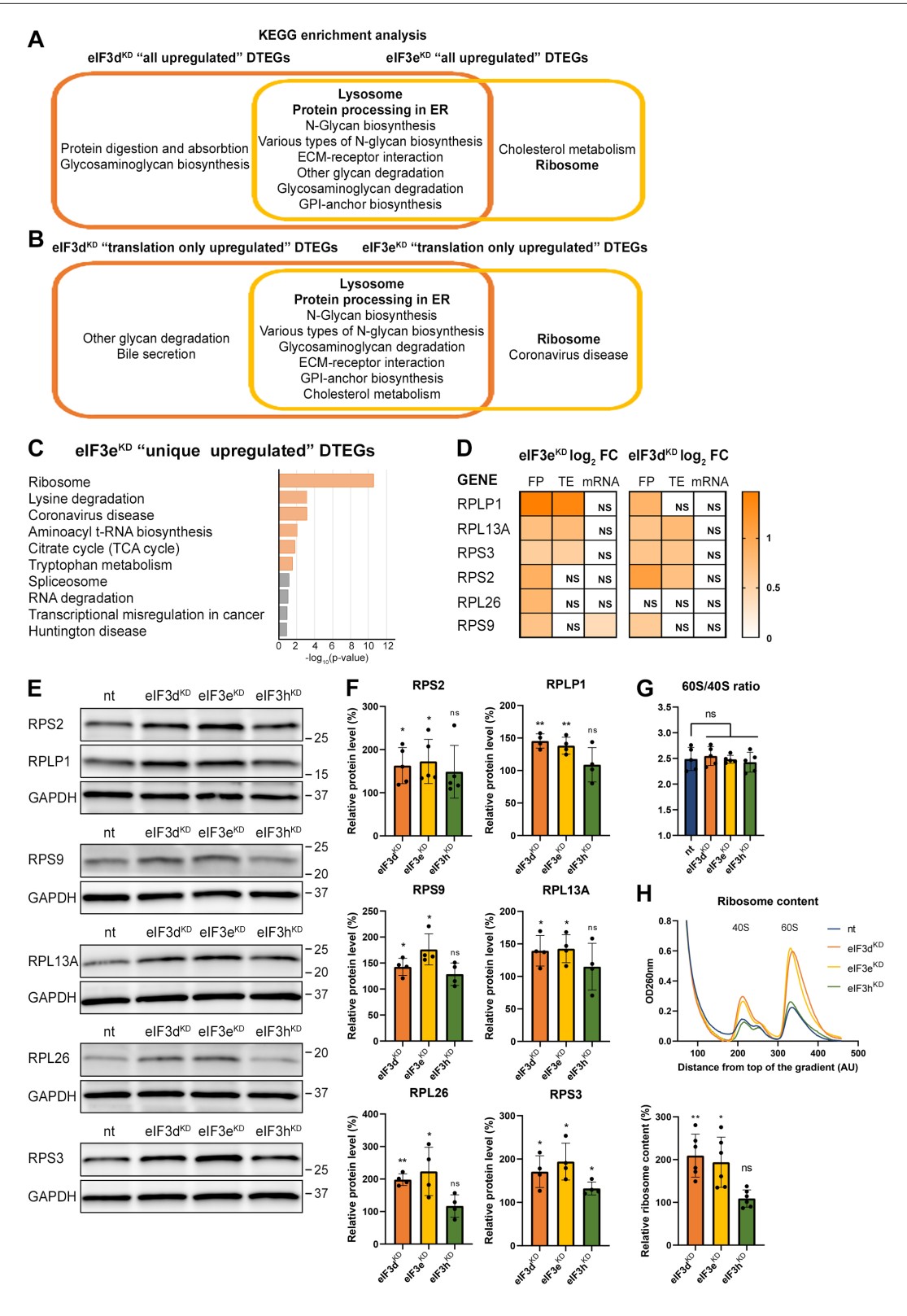

**Figure 3.** KEGG pathway enrichment analysis of the eIF3d[KD]- and eIF3e[KD]-associated upregulated DTEGs reveals upregulation of ribosomal proteins. (**A**) Venn diagram of the 10 most significantly enriched KEGG pathways for eIF3d[KD] 'all' and eIF3e[KD] 'all' groups of upregulated DTEGs, highlighting that most of the pathways are common to both knock-downs. Lysosome and Protein processing in ER are highlighted in bold. The complete results of the KEGG enrichment analysis with corresponding p-values can be found in *Figure 3—figure supplement 1A*. (**B**) Venn diagram as in A but displaying

*Figure 3 continued on next page*

*Figure 3 continued*

eIF3d[KD] 'translation only' and eIF3e[KD] 'translation only' groups of the upregulated DTEGs. (**C**) The bar chart shows the top 10 enriched KEGG terms for eIF3e[KD] 'unique' upregulated DTEGs. Orange bars correspond to terms with significant p-values (<0.05), grey bars correspond to terms with not significant p-values (>0.05). The p-values were calculated by Enrichr gene set search engine. (**D**) The list of genes pre-selected for western blot analysis and a heatmap showing their respective log2 fold-change values from differential expression analysis of FP, TE and mRNA in eIF3e[KD] and eIF3d[KD]. Positive values indicating significant upregulation are in shades of orange. ns = not significant p-adjusted value. (**E**) Western blot analysis of selected ribosomal proteins performed in the indicated knock-downs and the NT control. GAPDH was used as a loading control. (**F**) Relative protein levels of selected ribosomal proteins normalized to GAPDH; plots show mean ± SD, NT control = 100%. Dots represent results from individual biological replicates. Shapiro-Wilk test was used to test for normal distribution. One sample t-test was used for statistical evaluation, p-values: *=p < 0.05, **=p < 0.01, ns = not significant. (**G**) eIF3d[KD], eIF3e[KD], and eIF3h[KD] do not influence the balanced ribosomal subunits production. The 60 S/40 S ratio was calculated from polysome profiles carried out in the presence of 50 mM EDTA. Dots represent results from individual biological replicates. Plots show mean ± SD. Shapiro-Wilk test was used to test for normal distribution. Paired t test was used for statistical evaluation, all downregulations were individually compared to NT. ns = not significant p-value. (**H**) Ribosomal content is increased in eIF3d[KD] and eIF3e[KD]. One representative polysome profile, carried out in the presence of 50 mM EDTA, made from 10 million cells is shown in the upper panel. Relative ribosomal content normalized to NT control set to 100% is shown in the lower panel. Plot shows mean ± SD. Individual biological replicates are depicted as dots. Shapiro-Wilk test was used to test for normal distribution. One sample t-test was used for statistical evaluation, p-values: *=p < 0.05, **=p < 0.01. All plots in (**F–H**) were created in GraphPad Prism version 8.4.3 for Windows.

The online version of this article includes the following source data and figure supplement(s) for figure 3:

**Source data 1.** Original files for western blot analysis displayed in *Figure 3E*.

**Source data 2.** PDF file containing original western blots for *Figure 3E*, indicating the relevant bands and treatments.

**Figure supplement 1.** KEGG pathway enrichment analysis of the eIF3dKD- and eIF3eKD-associated upregulated DTEGs reveals upregulation of ribosomal proteins.

ratio remained unaffected (*Figure 3G*). Furthermore, we performed a similar experiment with exactly 10 million cells *per* each sample and confirmed that the total amount of ribosomes in cells is indeed increased in eIF3e[KD] and eIF3d[KD] (*Figure 3H*).

Overall, our results suggest that eIF3e and, to some extent, its binding partner eIF3d maintain control over production of RPs in the cell. A similar role has recently been attributed to eIF3k, which has been shown to specifically repress the synthesis of the small ribosomal protein RPS15A, and cells depleted of eIF3k showed increased ribosomal content (*Duan et al., 2023*). As mentioned above, targeting eIF3e with siRNA results in co-downregulation of not only eIF3e and eIF3d, but also of eIF3k and eIF3l (*Wagner et al., 2016*) which may at least partially explain these findings.

In addition to mitochondrial RPs, we also observed increased TE for several other mitochondrial proteins from the Complex I, Complex IV and Complex V (KEGG term Oxidative phosphorylation hsa00190; rank 17; #of genes 12 out of 134; p=0.0003431) in eIF3e[KD], but many of these transcripts, mainly from the Complex I, are encoded by mitochondrial DNA and translated in mitochondria directly, so these results should be verified by Mitoribosome profiling, as the traditional Ribo-Seq method is not suitable for this (*Pearce et al., 2021*).

Because RPs and other components of the translational machinery frequently harbor 5' terminal oligopyrimidine (TOP) motifs that render their translation highly sensitive to mTORC1 (*Philippe et al., 2020*; *Stolovich et al., 2002*), we first investigated whether mTORC1 translation is affected in eIF3d[KD] or eIF3e[KD] but found no significant change in TE for any mTORC1 components. Next, we examined the effect of depletion of eIF3 subunits on translation of all known TOP mRNAs. We observed that transcripts harboring TOP motifs (*Philippe et al., 2020*) were significantly translationally activated in eIF3d[KD] and eIF3e[KD], but not in eIF3h[KD] (*Figure 4A–C*). Interestingly, the effect of eIF3d[KD] and eIF3e[KD] on translation of non-TOP mRNAs that are mTORC1-sensitive (*Gandin et al., 2016*) differed from what was observed for TOP mRNAs, as non-TOP transcripts were predominantly translationally offset (buffered), such that translation efficiency was increased to maintain a constant level of poly-some association despite decreased total mRNA abundance (*Figure 4—figure supplement 1A–C*). Together, these findings suggest that eIF3e and eIF3d may play a role in tempering translation of TOP mRNAs under normal cellular conditions, and that the loss of these eIF3 subunits has distinct impact on the translatome as compared to mTORC1.

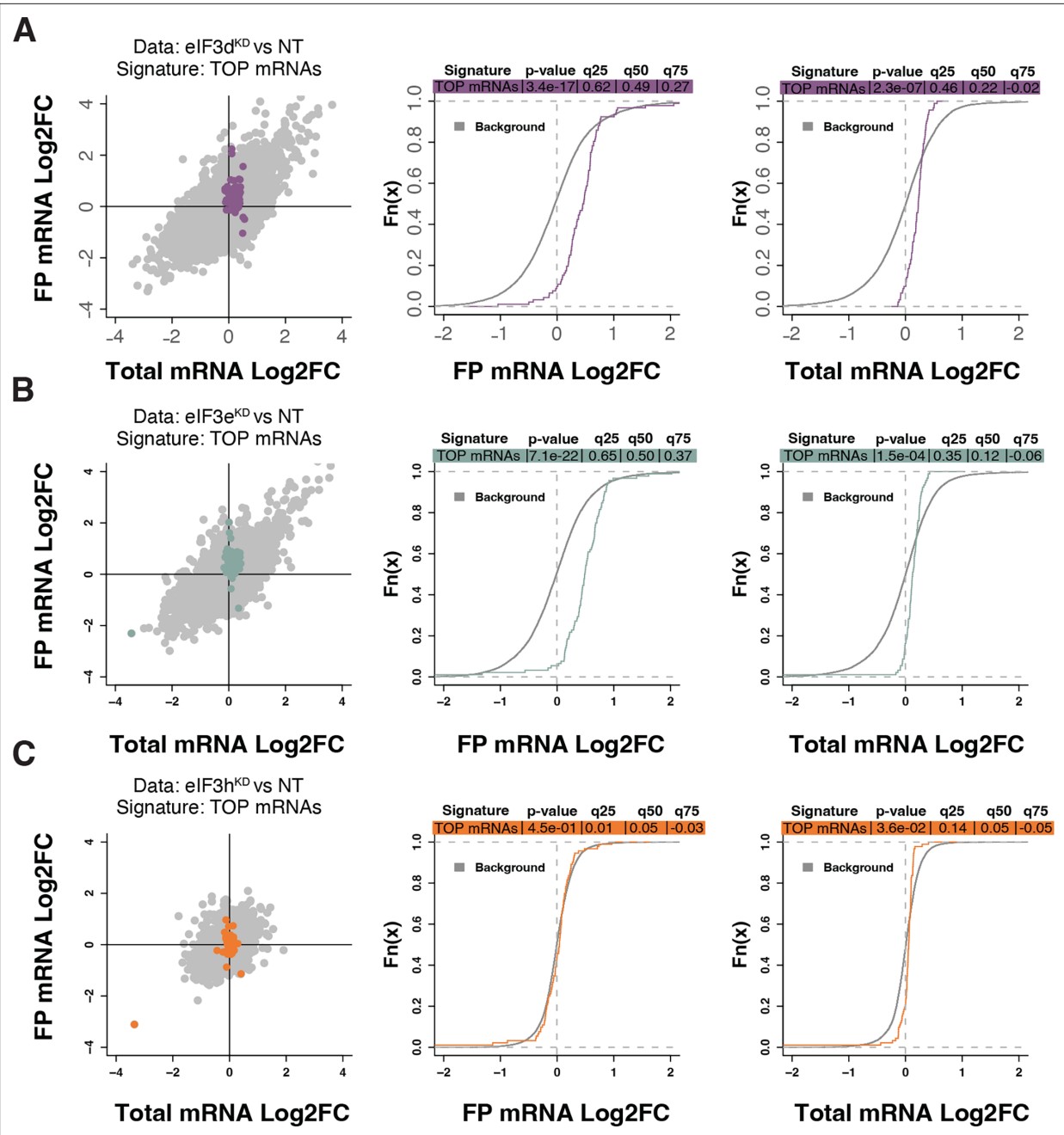

**Figure 4.** Loss of eIF3 subunits leads to translational activation of mRNAs with 5'UTR TOP motifs. (**A–C**) Scatterplots from translatome analysis of eIF3d[KD] (**A**), eIF3e[KD] (**B**), and eIF3h[KD] (**C**) with the location of transcripts harboring 5' UTR TOP motifs (*Philippe et al., 2020*) colored (left panels). Middle and right panels show the empirical cumulative distribution functions of log$_2$ fold changes in FP and total mRNA for the transcripts with TOP motifs. The background constituting all other transcripts are shown as grey curves. Significant differences between the distributions were identified using the Wilcoxon rank-sum test. Differences between the distributions at each quantile are indicated. Shift to the right indicates increased expression, while shift to the left indicates decreased expression. eIF3d[KD] and eIF3e[KD] show very significant increase of FPs of TOP mRNAs, suggesting mainly translational upregulation.

The online version of this article includes the following figure supplement(s) for figure 4:

**Figure supplement 1.** mTOR-sensitive transcripts tend to be translationally offset (buffered) with loss of eIF3d and eIF3e.

## eIF3d^KD and eIF3e^KD decrease translation efficiency of the key components of the MAPK/ERK and other signaling pathways implicated in cancer

The eIF3d^KD or eIF3e^KD 'all downregulated' and eIF3d^KD/eIF3e^KD 'common downregulated' groups were enriched for a similar set of KEGG pathways (*Figure 5A* and *Figure 5—figure supplement 1A*). The effect is generally stronger for eIF3d^KD, which simply means that this knock-down has a higher number of DTEGs in the same pathway than eIF3e^KD (for example 'MAPK signaling pathway' has 42 DTEGs in eIF3d^KD vs. 30 DTEGs in eIF3e^KD). Therefore, we focused primarily on DTEGs identified in eIF3d^KD 'all' (*Figure 5A* and *Figure 5—figure supplement 1A*). Most of the top 10 KEGG results are related to and/or encompass the 'MAPK signaling pathway' term (*Figure 5—figure supplement 1B*), like 'Chronic myeloid leukemia' and the 'Neurotrophin signaling pathway'. Depletion of eIF3d affected multiple components of the MAPK signaling pathways, which plays a critical role in the regulation of cell proliferation (*Zhang and Liu, 2002*) and cancer development and progression (reviewed in *Dhillon et al., 2007*). These include the docking protein Grb2 (encoded by *GRB2*) and guanine nucleotide exchange factor Sos (*SOS1*), through kinases c-Raf (*RAF1*), MEK1 (*MAP2K1*), Erk2 (*MAPK1*), MNK1 (*MKNK1*), Rsk2 (*RPS6KA3*), to the transcription factor CREB (*ATF4*). In addition, Rac1 and PAK1, that activate c-Raf by increasing phosphorylation of Ser338 (*King et al., 1998*) were also significantly affected (*Figure 1A*, *Figure 5—figure supplement 1C* and *Supplementary file 1*). Consistent with the observed translational suppression of MNK1, which phosphorylates eIF4E (*Waskiewicz et al., 1997*), mRNAs that critically depend on phospho-eIF4E levels (*Karampelias et al., 2022*) were translationally offset in eIF3d^KD and eIF3e^KD cells, such that translation efficiency was decreased to maintain a constant level of polysome association despite increased total mRNA abundance (*Figure 5—figure supplement 2A, B*).

Consistently, Gene Ontology (GO) analysis revealed strong association of downregulated DTEGs with biological processes like 'protein phosphorylation' and 'regulation of transcription' and the molecular function term 'protein serine/threonine kinase activity' (*Figure 5—figure supplement 3A, B*). Key integrin-mediated signaling proteins that were implicated in cancer cell plasticity, adhesion, migration, and survival (reviewed in *Cooper and Giancotti, 2019*) also had significantly decreased TE, specifically, proto-oncogenes Crk (*CRK*) and Crk-L (*CRKL*), *PRKCA* – a catalytic subunit of PI3-kinase, Rac1 GTPase (*RAC1*), JNK2 (*MAPK9*), as well as Abl1 kinase (*ABL1*) and BCR activator of Rho-GEF (*BCR*). The TGF-beta signaling, important for a wide range of cellular processes as well as for tumor cell invasion, metastasis, and resistance to chemotherapy (reviewed in *Colak and Ten Dijke, 2017*), was affected at the level of the intracellular transducer Smad4 (*SMAD4*) and receptor activated SMADs, namely Smad3 (*SMAD3*) and Smad5 (*SMAD5*) and TGF beta (*TGFB3*) itself. The NFkB pathway, a regulator of immune and inflammatory responses (*Liu et al., 2017*), also implicated in cancer cell proliferation, survival, angiogenesis and plasticity (reviewed in *Xia et al., 2014*), was downregulated at the level of NFkB (*NFKB1*) itself and IKK kinase (*CHUK*). Notably, a direct link between activation of NFkB and eIF3 has very recently been described (*She et al., 2023*). Finally, the canonical Wnt signaling pathway, which shapes the stem cells-like properties of cancer cells and boosts proliferation (reviewed in *Zhang and Wang, 2020*), was downregulated at the level of a protein Dishevelled (*DVL3*) and GSK3 beta kinase (*GSK3B*). Physiological relevance of all these hits remains to be seen; for the complete list of all DTEGs, please see *Supplementary file 1*.

## eIF3d^KD and eIF3e^KD reduce protein levels of key components of the activated MAPK/ERK pathway

Interestingly, more than half (26 of the 42) downregulated DTEGs belonging to the 'MAPK signaling pathway' can be classified as translationally buffered; that is, their mRNA levels increase while their TE decreases (*Kusnadi et al., 2022*). However, this is not a typical case of translational buffering because mRNA upregulation is not only nullified by translational repression, but completely overruled, so that the total number of FPs is also reduced. To verify that the reduction in TE and FP counts is indeed manifested by a reduction in cellular protein levels, we selected a set of genes with different combinations of changes in TE or FP or mRNA levels, including translationally exclusive and translationally buffered genes, as well as genes with unchanged TE, and subjected them to western blotting (*Figure 5B, D*).

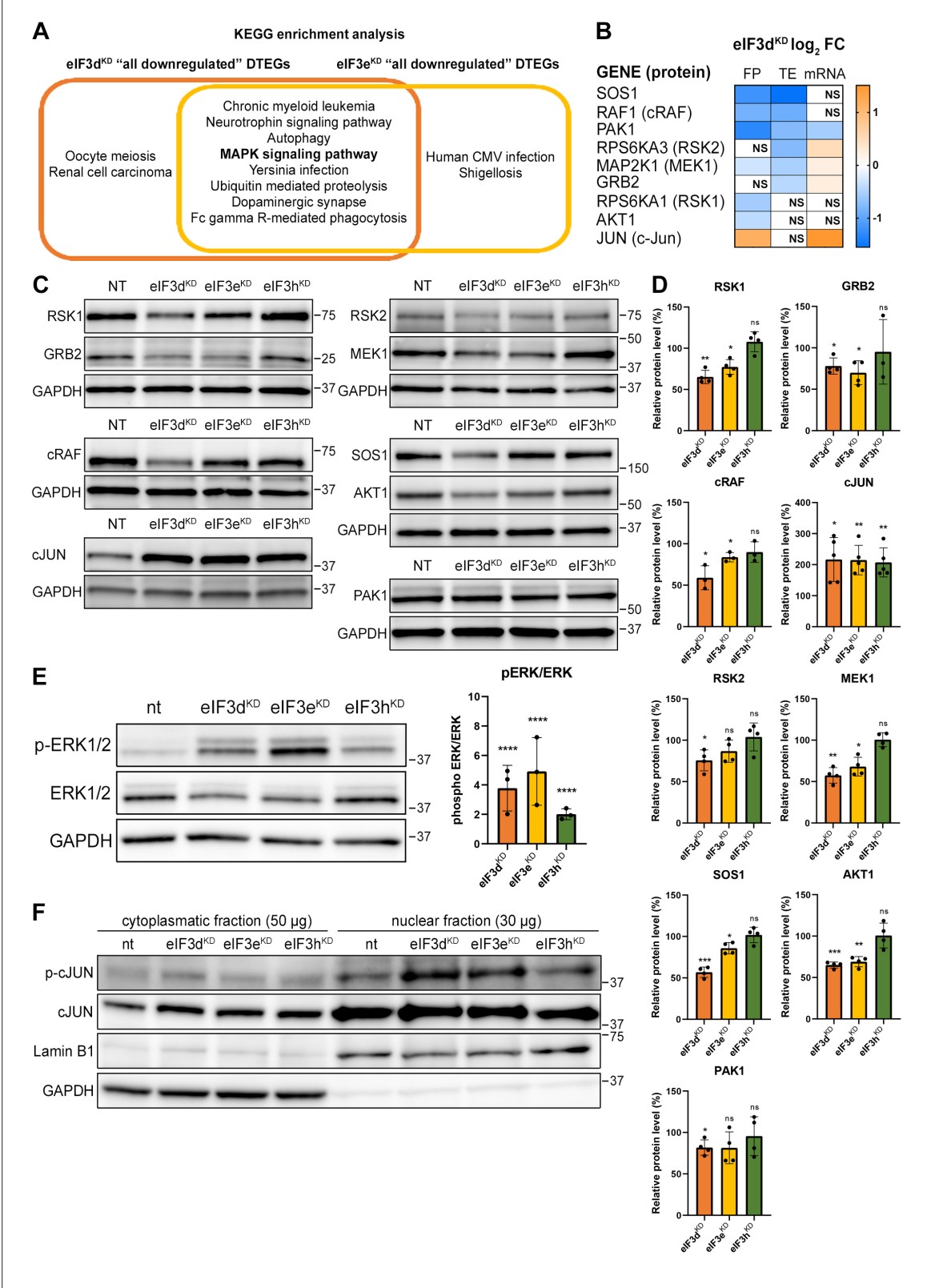

**Figure 5.** KEGG pathway enrichment analysis of the eIF3d^KD- and eIF3e^KD-associated downregulated DTEGs reveals downregulation of MAPK signaling pathways components. (**A**) Venn diagram of the 10 most significantly enriched KEGG pathways for eIF3d^KD 'all' and eIF3e^KD 'all' groups of downregulated DTEGs, highlighting that most of the pathways are common to both knock-downs. The MAPK signaling pathway is highlighted in bold. The complete results of the KEGG enrichment analysis with corresponding *p*-values can be found in ***Figure 5—figure supplement 1A***. (**B**) The list of

*Figure 5 continued on next page*

*Figure 5 continued*

genes pre-selected for western blot analysis and a heatmap showing their respective log2 fold-change values from differential expression analysis of FP, TE and mRNA in eIF3d^KD. Negative values indicating a significant downregulation are in shades of blue, while positive values showing significant upregulation are in shades of orange. ns = not significant p-adjusted value. (**C**) Western blot analysis of selected proteins constituting the MAPK/ERK pathway performed in the indicated knock-downs and the NT control. GAPDH was used as a loading control. (**D**) Relative protein levels of selected MAPK/ERK pathway proteins normalized to GAPDH; plots show mean ± SD, NT control = 100%. Dots represent results from individual biological replicates. Shapiro-Wilk test was used to test for normal distribution. One sample t-test was used for statistical evaluation, p-values: *=p < 0.05, **=p < 0.01, ***=p < 0.001, ns = not significant. (**E**) Western blot analysis of the phosphorylation status of the ERK1/2 proteins (left) and its relative quantification normalized to NT = 1 (right). Plot shows mean ± SD. Dots represent results from individual biological replicates. Shapiro-Wilk test was used to test for normal distribution. One-sample t-test was used for statistical evaluation, ****=p < 0.0001. (**F**) Western blot analysis of the phosphorylation status of the c-Jun transcription factor in cytoplasmatic and nuclear fractions in the indicated knock-downs and the NT control. The protein loading was 50 μg of total protein for cytoplasmatic lysate and 30 μg of total protein for nuclear lysate, as indicated. GAPDH was used as cytoplasmatic loading control and Lamin B1 was used as nuclear loading control. This experiment was repeated three times with similar results. All plots in (**D, E**) were created in GraphPad Prism version 8.4.3 for Windows.

The online version of this article includes the following source data and figure supplement(s) for figure 5:

**Source data 1.** Original files for western blot analysis displayed in *Figure 5C, E and F*.

**Source data 2.** PDF file containing original western blots for *Figure 5C, E and F*, indicating the relevant bands and treatments.

**Figure supplement 1.** KEGG pathway enrichment analysis of the eIF3d^KD- and eIF3e^KD-associated downregulated DTEGs reveals downregulation of 'MAPK signaling pathway' components.

**Figure supplement 2.** Loss of eIF3 subunits leads to translational offsetting of transcripts with enhanced translation downstream of phosphorylated eIF4E.

**Figure supplement 3.** GO enrichment analysis for eIF3d^KD associated downregulated DTEGs.

We found, that AKT1, GRB2, MAP2K1 (MEK1), PAK1, RPS6KA1 (RSK1), RPS6KA3 (RSK2), RAF1 and SOS1 had visibly decreased protein levels in eIF3d^KD and most of them also in eIF3e^KD. An interesting exception is the transcription factor c-Jun, which showed increased mRNA and FP levels, unchanged TE, but robust increases in protein levels in all three knock-downs (*Figure 5C, D*). The mRNA levels of all selected genes were verified by qPCR, showing a good correlation with our sequencing data; whereas PAK1 showed significant decrease, JUN mRNA was significantly increased in both eIF3d^KD and eIF3e^KD (*Figure 5—figure supplement 3C*). To assess the functional consequences of the down-regulation of MAPK/ERK pathway components, we examined the phosphorylation status of effector kinases ERK1/2, which directly phosphorylate several transcription factors including c-Jun (*Leppä et al., 1998*). As with other MAPK proteins, ERK1/2 expression was reduced in eIF3d^KD and eIF3e^KD, but its phosphorylation at activation sites Thr202/Tyr204 was unexpectedly increased in all three knock-downs (*Figure 5E*). Next, we examined the phosphorylation status of the upregulated c-Jun protein. As phosphorylated c-Jun translocates to nucleus to activate transcription of specific genes, we performed western blot of both cytoplasmatic and nuclear fractions. In accord with increased ERK1/2 phosphorylation, c-Jun was found to be activated by phosphorylation at Ser63 and largely accumulated in the nucleus (*Figure 5F*). Thus, despite reduced MAPK/ERK pathway protein levels, MAPK/ERK signaling appears to be activated when eIF3 function in translation is compromised.

## Intact eIF3 contributes to translational regulation of ATF4 and MDM2

Knocking down eIF3h generates an eIF3 subcomplex lacking eIF3h and two nonessential subunits eIF3k and eIF3l (*Figure 1B*), having only negligible impact on overall translation. Accordingly, we identified 19 transcripts whose TE was decreased and only 2 transcripts whose TE was increased compared to the NT control (*Figure 2D*). Given the low number of DTEGs, we observed no significant enrichment of any KEGG pathway within the upregulated DTEGs and only a few significant enrichments for the eIF3h^KD 'all downregulated' DTEGs in pathways like 'Prostate cancer', 'Viral carcinogenesis' or 'PI3-Akt signaling pathway', all of which included only two common hits – MDM2 and ATF4 (*Figure 6A, Supplementary file 2A*). Additionally, we also noticed enrichment of GO terms associated with ubiquitination (*Figure 6B, Supplementary file 2B*), which was mainly due to the MDM2, WAC, RNF10, RMND5A, USP38, and ZRANB1 genes, all of which showed decreased TE. MDM2, WAC, RNF10 and RMND5A are all E3 ubiquitin ligases while USP38 and ZRANB1 are de-ubiquitinases. Note that all of the above downregulated DTEGs are common for all three knock-downs.

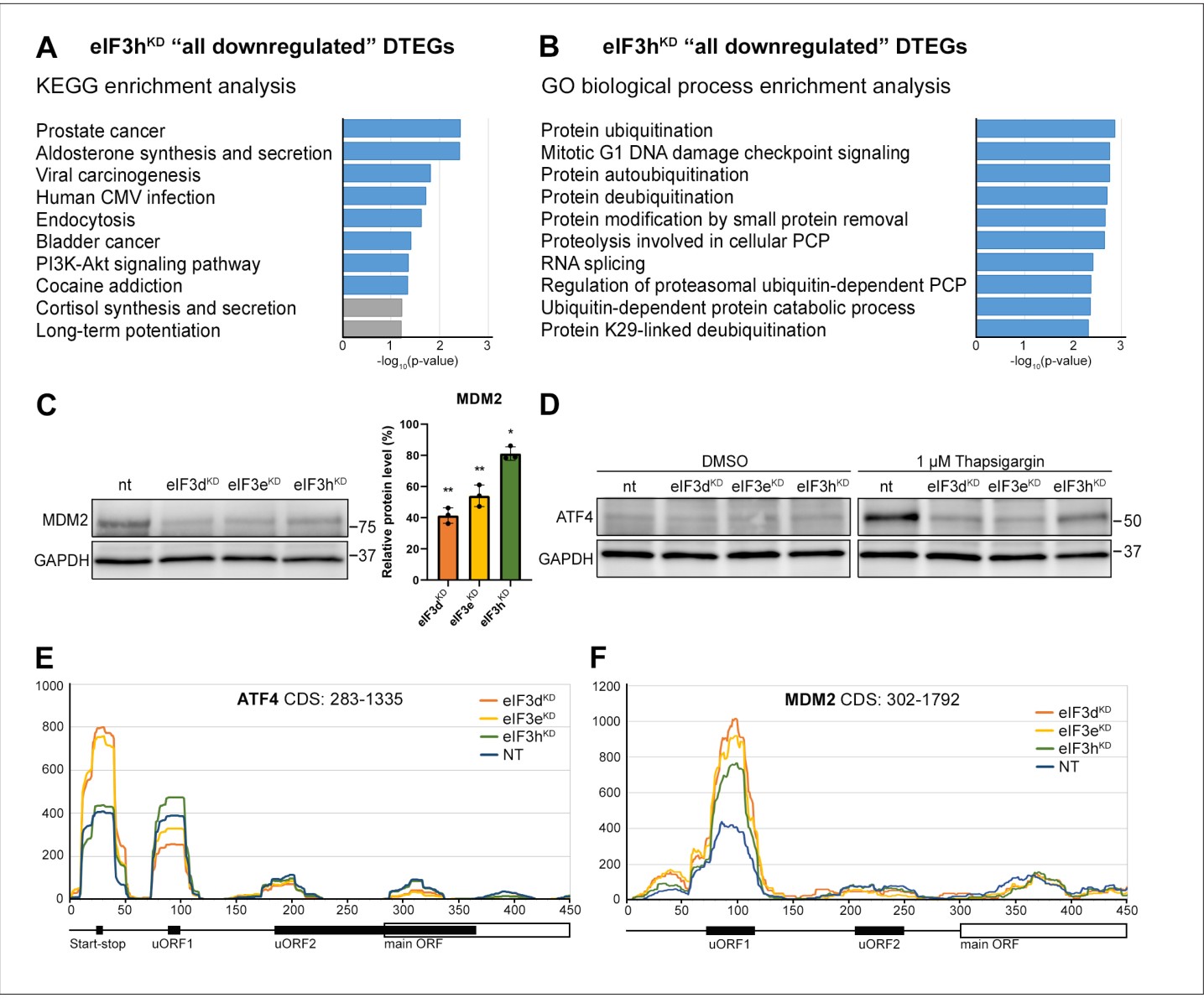

**Figure 6.** KEGG pathway and GO enrichment analysis of the eIF3h^KD-associated DTEGs reveals downregulation of proto-oncogene MDM2 and a defective stress-induced upregulation of ATF4. (**A**) The bar chart shows the top 10 enriched KEGG terms for eIF3h^KD 'all downregulated' DTEGs. Blue bars correspond to terms with significant p-values (<0.05), grey bars correspond to terms with not significant p-values (>0.05). The p-values were calculated by Enrichr gene set search engine. (**B**) The bar chart shows the top 10 enriched GO Biological Process terms for eIF3h^KD 'all downregulated' DTEGs. Blue bars correspond to terms with significant p-values (<0.05). The *p*-values were calculated by Enrichr gene set search engine. PCP; protein catabolic process. (**C**) Western blot analysis of the MDM2 expression preformed in the indicated knock-downs and NT control. GAPDH was used as a loading control; plot shows mean ± SD, NT control = 100%. Dots represent results from individual biological replicates. Shapiro-Wilk test was used to test for normal distribution. One sample t-test was used for statistical evaluation, p-values: *=p < 0.05, **=p < 0.01. Plots was created in GraphPad Prism version 8.4.3 for Windows. (**D**) Western blot analysis of the stress-induced upregulation of ATF4 expression performed in the indicated knock-downs and NT control. Before harvesting, cells were incubated for 3 hr with 1 μM Thapsigargin to induce ER stress or with DMSO as a stress-free control. GAPDH was used as a loading control. (**E**) Normalized ribosomal footprint coverage along the *ATF4* mRNA (first 450 nucleotides). Schematic with regulatory elements is shown at the bottom. Average of all three replicates is shown. Footprint coverage was normalized to all footprints mapping to the *ATF4* mRNA. (**F**) Same as in (**E**) only for *MDM2* mRNA.

The online version of this article includes the following source data and figure supplement(s) for figure 6:

**Source data 1.** Original files for western blot analysis displayed in *Figure 6C and D*.

**Source data 2.** PDF file containing original western blots for *Figure 6C and D*, indicating the relevant bands and treatments.

**Figure supplement 1.** The *ATF4* mRNA is upregulated in all three eIF3 knock-downs tested.

**Figure supplement 2.** Loss of eIF3 subunits modulates signatures of DAP5-dependent translation.

Perhaps the most interesting of these genes is the proto-oncogene MDM2, an E3 ubiquitin ligase that acts as a negative regulator of the p53 protein (*Marine and Lozano, 2010*), whose overexpression has been reported in many different cancer types (*Rayburn et al., 2005*). We did confirm that the MDM2 protein levels were reduced in all three knock-downs (*Figure 6C*).

An equally interesting DTEG is ATF4, which is among the top 10 most significant DTEGs in all three knock-downs (*Figure 2B, D*). In eIF3h[KD], it can be classified as a translationally buffered gene because it has strongly upregulated mRNA levels, but this increase is not accompanied by a concomitant increase in the FP count, thereby resulting in decreased TE. In eIF3d[KD] and eIF3e[KD], it is even significantly translationally repressed (FPs are decreased). Using qPCR, we confirmed ATF4 mRNA upregulation in all three knock-downs (*Figure 6—figure supplement 1A*). Because ATF4 protein levels are on the verge of the western blot detection limit under non-stress conditions, we examined whether all three knock-downs tested had any effect on ATF4 induction upon 3 hour-long ER stress provoked by 1 μM Thapsigargin (*Harding et al., 2000*). Indeed, stress-mediated ATF4 induction was almost abolished in eIF3d[KD] and eIF3e[KD], and strongly decreased in eIF3h[KD] (*Figure 6D*). This differential effect correlates well with our sequencing data, where the decrease in TE was also stronger in eIF3d[KD] and eIF3e[KD]. The ATF4 protein levels in DMSO control are shown as a no stress reference. Overall, these findings further confirm that an intact eIF3 complex is required for proper translational control of ATF4, as we and others have previously suggested (*Guan et al., 2017*; *Hronová et al., 2017*; *Mukhopadhyay et al., 2023*; *Shu et al., 2022*; *Wagner et al., 2020*), and as discussed below.

As both *ATF4* and *MDM2* are known to contain uORFs in their 5' UTR that regulate their translation under stress vs. non-stress conditions (*Akulich et al., 2019*; *Dey et al., 2010*; *Lu et al., 2004*; *Smirnova et al., 2024*; *Vattem and Wek, 2004*), we examined the footprint coverage in the 5' UTR of these genes in more detail. In the case of *ATF4*, a markedly increased accumulation of footprints was observed at the most 5' Start-stop (St-st) element, specifically in eIF3d[KD] and eIF3e[KD], while coverage of uORF1 decreased, and so did also the basal level of uORF2 (modestly) and *ATF4* (*Figure 6E*). This agrees well with our observations that ribosomes are stalled at this St-st element (*Rendleman et al., 2023*; *Wagner et al., 2020*) and further suggests that both eIF3e and eIF3d are required to clear these St-st-trapped ribosomes. Regarding *MDM2*, footprint coverage was increased specifically for uORF1 in all three knock-downs (*Figure 6F*), consistent with its several-fold stronger inhibitory role on MDM2 expression compared to uORF2 (*Jin et al., 2003*), which seems to be magnified in the absence of fully functional eIF3. Collectively, these results are in line with the well-established role of eIF3 in translation reinitiation (reviewed in *Valášek et al., 2017*). In addition, we found similarly accumulated footprints also at uORF1 and uORF2 of the RAF1 kinase, one of the downregulated components of the MAPK pathway (*Figure 6—figure supplement 1B*).

Last but not least, eIF3d was shown to mediate mechanism of alternative cap-dependent translation initiation in conjunction with DAP5 (p97/NAT1) (*Alard et al., 2023*; *de la Parra et al., 2018a*; *Volta et al., 2021*) and, importantly, DAP5-dependent mRNAs also frequently contain uORFs (*David et al., 2022*). Consistent with this, the effects of depletion of DAP5 on the translatome were comparable to those observed in eIF3d-depleted cells (*Figure 6—figure supplement 2A–C*); that is, no changes in mRNA levels and the FP coverage moving in the same direction, either up or down. Since similar effects were also observed with eIF3e[KD] and eIF3h[KD], it seems that the DAP5-dependent mechanism may rely upon not only eIF3d but also on the entire eIF3 complex.

## Presence of uORFs, UTR length and GC content correlate with the change in translation efficiency

eIF3 is critically required for mRNA recruitment to the 40 S ribosomal subunit (reviewed in *Valášek et al., 2017*) and several eIF3 subunits have been proposed to regulate translation of specific subsets of mRNAs (*Lee et al., 2015*; *Lee et al., 2016*). We therefore asked whether the changes in TE that were observed in the eIF3d, eIF3e or eIF3h knock-downs correlate with specific mRNA properties. Indeed, in all three knock-downs, the change in TE was negatively correlated with the length of 5' and 3' UTR, and with the presence of uORFs in 5' UTR (Spearman correlation, $p < 1 \times 10^{-45}$) (*Figure 7A*). Moreover, we found a significant positive Spearman correlation between TE changes and the GC content in the coding region (CDS) and 3' UTRs.

To explore these correlations in more detail, we focused only on those mRNAs that are exclusively regulated at the translational level in each knock-down and therefore are more likely to contain

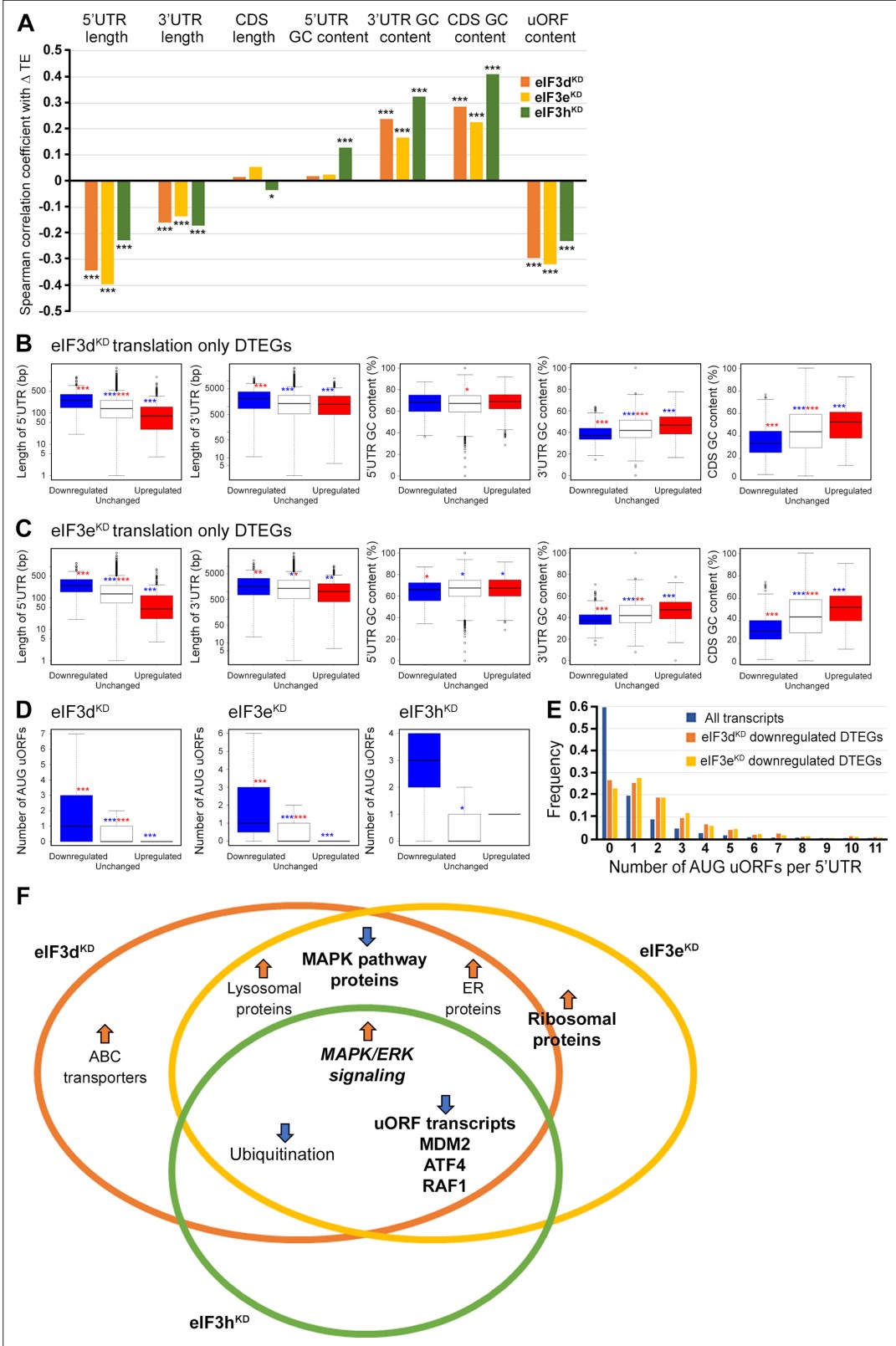

**Figure 7.** Differential TE transcripts in eIF3d^KD and eIF3e^KD show negative correlation with the UTR length and uORF content while they positively correlate with GC content of their 3' UTRs and coding sequences (CDS). (**A**) Bar plot showing the Spearman correlation between the observed ΔTE values for all genes with assigned adjusted *p*-values in each knock-down and different mRNA features (5' and 3' UTR length or GC content, uORF content

*Figure 7 continued on next page*

*Figure 7 continued*

in 5' UTR). ***=p < 10⁻²⁰, **=p < 10⁻¹⁰, *=p < 0.005. (**B, C**) Box and whisker plots comparing UTR lengths or GC content (in UTRs or CDS) among mRNAs with TE significantly increased (red, upregulated), decreased (blue, downregulated), or unchanged (white) in eIF3d$^{KD}$ (**B**) and eIF3e$^{KD}$ (**C**); ***=Padj < 10⁻¹⁰, **=Padj < 10⁻⁵, *=Padj < 0.05, color indicates comparison set. (**D**) Same as in (**B, C**) but for the number of AUG-initiated uORFs in 5' UTRs in the eIF3d, eIF3e, and eIF3h knock-downs. Outliers are not shown for better clarity. (**E**) Histogram of the frequency of AUG-initiated uORFs per 5' UTR in all transcripts listed in the uORFdb database that were assigned a *p*-value in this study (n=11450), for downregulated translation only DTEGs in eIF3d$^{KD}$ (n=1027) and for downregulated translation only DTEGs in eIF3e$^{KD}$ (n=618). (**F**) Venn diagram summarizing the main results of this study. Groups of significant DTEGs identified in given knock-downs (encircled – color-coded) are indicated by a blue arrow for downregulated or an orange arrow for upregulated. Results confirmed by western blotting are shown in bold. Ribosomal proteins were placed on the borderline between eIF3d$^{KD}$ and eIF3e$^{KD}$ because they were identified as eIF3e$^{KD}$ 'unique upregulated' DTEGs, but their protein levels were elevated in both eIF3e$^{KD}$ and eIF3d$^{KD}$, as shown by western blotting. The 'MAPK/ERK signaling' is shown in bold italics because it is an independent phenomenon that could be detected only by western blotting.

The online version of this article includes the following figure supplement(s) for figure 7:

**Figure supplement 1.** eIF3dKD, eIF3eKD, and eIF3hKD do not show accumulation of footprints at the beginning of CDSes, as reported previously, but display a significant overlap with mRNAs directly interacting with eIF3.

---

distinct features that facilitate their translational regulation. We divided them into those with significantly changed TE (decreased and increased) and those with unchanged TE and searched for significant differences in the median values of the tested traits (*Figure 7B, C*). In accordance with the Spearman correlation, in eIF3d$^{KD}$ and eIF3e$^{KD}$, mRNAs of downregulated DTEGs had longer 5' UTRs, while upregulated DTEGs possessed shorter 5' UTRs, when compared to mRNAs with unchanged TE. The same trend is seen for the 3' UTR length in eIF3e$^{KD}$, but with lower significance. In case of eIF3d$^{KD}$, only the downregulated DTEGs showed a significant change in 3' UTR length, but not DTEGs that were upregulated.

In terms of GC content, while we observed little significant difference in GC content in the 5' UTR of eIF3d$^{KD}$ and eIF3e$^{KD}$, there was a highly significant difference in the CDS and 3' UTR of these knock-downs in both downregulated and upregulated DTEGs, indicating that mRNAs with increased TE tend to have more GC-rich 3' UTRs and, surprisingly, CDS and vice versa. For eIF3h$^{KD}$, we only observed differences of low significance due to the low number of DTEGs that were only translationally regulated (only 8 downregulated and 1 upregulated) (data not shown).

In summary, mRNAs with decreased TE in cells wherein eIF3d and eIF3e were depleted exhibited on average longer 5' and 3' UTRs and lower GC content in the CDS and 3' UTRs. In turn, mRNAs with increased TE have on average shorter 5' UTRs and a higher GC content in the CDS and 3' UTRs.

We next asked whether 5' UTRs of upregulated or downregulated DTEGs contain any specific sequence motifs or short upstream open reading frames (uORFs). We did not identify any specific sequence motif, but expectedly, the 5' UTRs of 'downregulated translation only DTEGs have a strong tendency to contain uORFs starting with AUG (*Figure 7D, E*). These findings suggest that the intact eIF3 complex is primarily required for translational control of subset of mRNAs containing short uORFs in their 5' UTR even under normal cellular conditions, like for example *MDM2*, *ATF4* or *RAF1*, which is in line with its role in translation reinitiation (*Valášek et al., 2017*), and further discussed below.

## Discussion

In this study, we interrogated the effects of altering composition of eIF3 by depleting three eIF3 subunits in HeLa cells, followed by RNA-Seq and Ribo-Seq to elucidate their role in translational regulation of specific mRNAs by differential expression analysis. We focused on differential translation efficiency genes (DTEGs) and identified several distinct groups of DTEGs associated with individual eIF3 knock-downs.

Interestingly, while DTEGs identified in eIF3d$^{KD}$ and eIF3e$^{KD}$ largely overlapped, considerably more DTEGs were identified in eIF3d$^{KD}$ then in eIF3e$^{KD}$, correlating well with the severity of the growth phenotype of both knock-downs, with the former being more detrimental despite not affecting eIF3 integrity (*Wagner et al., 2016*).

Why it is so is currently unknown. Perhaps the simplest explanation is that the higher number of DTEGs identified in the same pathways in eIF3d^KD vs. eIF3e^KD originated in different eIF3d protein levels between the two knock-downs. Despite being markedly co-downregulated in eIF3e^KD, eIF3d levels were more dramatically reduced in eIF3d^KD compared to eIF3e^KD, as expected. Furthermore, eIF3d is a peripheral subunit that binds to the head of the 40 S subunit (mainly via the small ribosomal protein Rps16 *Herrmannová et al., 2020*) and is connected to the eIF3 octamer not only via eIF3e but also via the eIF3a and c subunits (*Bochler et al., 2020*). Therefore, we hypothesize that eIF3d may remain partially operational even in the absence of eIF3e, and therefore the more severe phenotype in eIF3d^KD is likely to stem from lower levels of the eIF3d protein compared to eIF3e^KD. Following this logic, phenotypes that are common to eIF3d^KD and eIF3e^KD are likely a consequence of reduction in eIF3d protein levels. Another way to explain this ostensible paradox is that eIF3d could have functions outside of the eIF3 complex. The fact that in the absence of eIF3e – its interaction partner, eIF3d gets rapidly degraded, however, speaks against this option. It is also possible that eIF3d, as a head-interacting subunit close to an important head-situated ribosomal protein RACK1, which interacts with eIF3c and serves as a landing pad for regulatory proteins (*Kouba et al., 2012*; *Nilsson et al., 2004*), could be a target of signaling pathways. This may render this eIF3 subunit critically important for translation of specific mRNAs. In support, eIF3d (in the context of entire eIF3) was shown to be regulated by stress-triggered phosphorylation (*Lamper et al., 2020*) and, together with DAP5, it was shown to promote translation by an alternate cap-dependent (eIF4F-independent) mechanism (*de la Parra et al., 2018a*; *Lee et al., 2016*).

In any case, perhaps the most intriguing finding is that eIF3d^KD and, to a large extent, also eIF3e^KD result in translational repression of a marked number of the MAPK pathways constituents, significantly reducing their protein levels, despite the apparent compensatory mechanisms that increase their mRNA levels. Somewhat unexpectedly, despite the downregulation of a number of its components, eIF3d^KD and eIF3e^KD but also eIF3h^KD appear to activate MAPK/ERK pathway as evidenced by the elevated ERK1/2 and c-Jun phosphorylation. Notably, the eIF3d influence on the MAPK/ERK pathway was previously demonstrated in HTR-8/SVneo cells, where overexpression of eIF3d led to hypo-phosphorylation of MEK1 and ERK1/2, while downregulation had the opposite effect; however, no significant effect on the protein levels of its components was detected in this study (*Li et al., 2021*). Importantly, since the increased ERK1/2 phosphorylation was observed not only in eIF3d^KD and eIF3e^KD but also in eIF3h^KD, where expression of MAPK pathways proteins is unaffected, we conclude that activation of the MAPK/ERK pathway is a general consequence of the compromised eIF3 function.

Expression of the transcription factor c-Jun is partially controlled by the MAPK/ERK pathway and, consistently, we found that it is strongly upregulated in all three knock-downs. Moreover, we observed that all three knockdowns activated c-Jun as evidenced by its increased nuclear translocation. Although *JUN* mRNA was previously shown to interact directly with eIF3 through a stable stem loop occurring in its 5' UTR, which positively stimulated its translation (*Lee et al., 2015*), in this study we observed upregulation of the *JUN* mRNA abundance, whereby the depletion of eIF3d, e, or h had no conspicuous effect on its translational efficiency. This can be explained by a recent report showing that eIF3d binds to *JUN* mRNA with increased affinity during chronic ER stress, suggesting that eIF3d-dependent translational regulation may be triggered by stress (*Mukhopadhyay et al., 2023*). Because c-Jun is a component of the AP1 transcription factor complex, wherein it either homodimerizes or heterodimerizes with other AP1 proteins, we examined the effects of eIF3 subunit depletion on mRNA, FP and TE trends of other AP1 family members. Similarly to c-Jun, only FOSL1 and ATF3 showed increased mRNA levels and FP counts with no net changes in TE, whereas other members of the JUN (JUNB and JUND) and FOS (FOS, FOSB, FOSL2) families remained unaltered. In addition, the ATF family members ATF2 and ATF4 were even downregulated (*Supplementary file 1*). Notwithstanding that our data suggest that impairment of eIF3 function may be compensated by the activation of the MAPK/ERK pathway, further research is required to unravel the full spectrum of changes associated with the MAPK pathways in eIF3-impaired cells. At the current state of our knowledge, we can only speculate that when translation is compromised, cells attempt to counteract it in two ways: (1) they produce more ribosomes to increase translational rates, and (2) activate MAPK signaling to send pro-growth signals that may ultimately further increase ribosome biogenesis, as further discusses below.

Recently, Lin et al. carried out Ribo-seq analysis in MCF-10A cells downregulated for eIF3e (*Lin et al., 2020*). They identified 240 downregulated and 220 upregulated eIF3e^KD-specific DTEGs that

overlap very poorly with our DTEGs (data not shown). In addition, they identified a set of 2683 eIF3e-dependent mRNAs encoding many proteins associated with mitochondrial and membrane functions that showed increased ribosome density between codons 25 and 75 in eIF3e$^{KD}$. Those mRNAs were therefore proposed to require eIF3e to recruit chaperons to promote their early elongation. While this set of genes displayed a significant overlap with our DTEGs (see below), we did not observe a similar phenomenon (a specifically increased ribosome density early on) in our datasets. In metagene analysis, we did observe some accumulation of footprints at the beginning of the CDS, but this was present in all samples, including NT, whereby peaks were located around codon 20 downstream of the translation initiation sites (*Figure 7—figure supplement 1A*).

Nonetheless, still considering that the 'accumulation effect' observed by Lin et al., most probably resulting from ribosome stalling and/or slower elongation rates, could artificially increase FPs and TE without actually increasing synthesis of the corresponding proteins in our datasets, we trimmed the first 225 nt (75 codons) from all mRNA's CDSs, and repeated the differential expression analysis in the same way. When compared, the two DE analyses showed approximately 80–90% identical DTEGs and high correlation (Spearman correlation coefficient of eIF3d$^{KD}$: 0.924, eIF3e$^{KD}$: 0.925, eIF3h$^{KD}$: 0.834), suggesting that our DE analysis is not biased by ribosome accumulation in the first 75 codons and reflects true changes in translation efficiency.

Despite this 'accumulation effect' discrepancy, comparing our DTEGs with the Lin et al. eIF3e-dependent data, we found a very significant overlap of 446 genes between the eIF3e-dependent mRNAs and our eIF3d$^{KD}$ 'all upregulated' dataset (p<1 × 10$^{-53}$), while the overlap with eIF3e$^{KD}$ 'all upregulated' – 251 genes – was surprisingly smaller (p<1 × 10$^{-24}$). Moreover, our KEGG enrichment analysis of eIF3e-dependent mRNAs from Lin et al. revealed the same top 2 KEGG pathways (data not shown) as in case of eIF3d$^{KD}$ and eIF3e$^{KD}$ 'all upregulated' groups; that is Lysosome and Protein processing in ER (*Figure 3A* and *Figure 3—figure supplement 1A*). Considering that eIF3d was co-depleted with eIF3e in the eIF3e$^{KD}$ of Lin et al., it is conceivable that not eIF3e but the eIF3d subunit (or cooperatively both subunits together) ensures loading of quality control factors to promote elongation of transmembrane proteins. In any case, resolving this issue will require further research.

Based on our analysis, eIF3e seems to be primarily involved in controlling the balanced production of mature ribosomes, as its loss resulted in significant number of 'unique upregulated' DTEGs with strong enrichment for the KEGG Ribosome and GO Translation and Ribosome biogenesis terms (*Figure 3C* and *Figure 3—figure supplement 1B*). Focusing directly on RPs, nearly twice as many RPs with significantly altered TEs were identified in eIF3e$^{KD}$ compared to eIF3d$^{KD}$ (32 vs 16) and a similar pattern was also evident at the FP level (91 vs 64). However, despite this obvious difference, RPs were increased similarly in both eIF3d$^{KD}$ and eIF3e$^{KD}$ (*Figure 3E, F and H*). This discrepancy between our Ribo-Seq and Western blot results in eIF3e$^{KD}$ *versus* eIF3d$^{KD}$ (*Figure 3D vs. E*) might be explained by the enhanced stabilization of RPs by the previously described compensatory dosage mechanism at the protein level (*Ishikawa, 2021*). In particular, to preserve the stoichiometry of the ribosome when numerous RPs are translationally upregulated, the rest of the RPs would be protected from degradation. This speculation can be supported by the impact of all knock-downs on translation efficiency of different E3 ubiquitin ligases, as described for eIF3h$^{KD}$ (*Supplementary file 2B*). Similar effect of increased RPs production was recently reported also for eIF3k$^{KD}$ (*Duan et al., 2023*). Of note, eIF3e$^{KD}$ also decreases protein levels of eIF3k and eIF3l and thus this RP-specific effect could be at least partially attributed to the loss of eIF3k. However, increased RP production was also observed in eIF3d$^{KD}$, where eIF3k is normally present, but was not observed in eIF3h$^{KD}$, where eIF3k and eIF3l are also co-depleted. Thus, although it now seems well-established that the eIF3 complex exerts control over RP production, the exact contribution of its individual subunits to this regulation awaits further analysis.

An important hint could be that all cytoplasmic RPs with increased TE in eIF3e$^{KD}$ (and most of mitochondrial RPs) were upregulated solely at the translational level (their mRNA levels did not change), which may well correspond to the fact that RP mRNAs belong to the TOP mRNAs family. These mRNAs contain a polypyrimidine tract in their 5' UTR and were suggested to be regulated solely at the translational level in response to stress or unfavorable growth conditions through mTORC1 or PI3-kinase pathways (*Cockman et al., 2020*; *Philippe et al., 2020*; *Stolovich et al., 2002*). Besides RPs, this family also includes some translational initiation and elongation factors (for the list of TOP mRNAs see *Supplementary file 1*). Accordingly, we found that TOP transcripts were significantly

translationally activated in eIF3d$^{KD}$ and eIF3e$^{KD}$ (*Figure 4A–C*), suggesting a role of eIF3e and eIF3d in repressing translation of TOP mRNAs under normal cellular conditions. Interestingly, since mTOR-activated translation of non-TOP transcripts was predominantly translationally offset (buffered) in eIF3d$^{KD}$ and eIF3e$^{KD}$ cells (*Figure 4—figure supplement 1A–C*), it appears that both effects are independent of each other; that is, mTOR does not generally operate *via* eIF3 and eIF3 does not require mTOR activation to fine tune RP production. It is also noteworthy that upregulation of cytosolic ribosomal proteins has previously been reported in association with the eIF3m downregulation – albeit only at transcriptional level (*Smekalova et al., 2020*), or with downregulation of eEF2 (*Gerashchenko et al., 2020*).

In addition, as we were unable to find any specific sequence motif in the 5' UTRs of the mRNAs whose TE was affected in eIF3 subunit-depleted cells, we cross-referenced our eIF3d$^{KD}$ 'all' and eIF3e$^{KD}$ 'all' groups with transcripts that physically interact with eIF3 as reported by *Lee et al., 2015*. We found that out of the 402 transcripts with a putative eIF3 binding sites in their 5' UTRs, 92 and 65 had significantly altered TE in the eIF3d$^{KD}$ and eIF3e$^{KD}$, respectively (51 were common for both eIF3d$^{KD}$ and eIF3e$^{KD}$; *Figure 7—figure supplement 1B*). This notable overlap involving for example the MAPK1 gene further underscores the robustness of our screening approach.

Among the few downregulated DTEGs in eIF3h$^{KD}$, we found *ATF4* and *MDM2* mRNAs, whose translation is tightly regulated by the presence of uORFs in their 5' UTRs in response to specific stress conditions (*Akulich et al., 2019*; *Dey et al., 2010*; *Jin et al., 2003*; *Smirnova et al., 2024*; *Vattem and Wek, 2004*). Here, we showed that these mRNAs were translationally downregulated in our knock-downs under non-stress conditions and, accordingly, their most 5' short uORFs (or St-st in the case of *ATF4*) accumulated FPs, most probably slowing down the flow of ribosomes downstream. The fact that transcripts of these genes contain uORFs in their 5' UTRs also correlates well with our finding that 5' UTRs of 'downregulated translation only' DTEGs have a strong tendency to contain AUG-initiated uORFs (*Figure 7D, E*). Noteworthy, the impaired *ATF4* induction in our eIF3 knock-downs under ER stress (*Figure 6D*), was reported before also for knock-downs of eIF3a, eIF3c, eIF3d, eIF3g, but not eIF3l (*Guan et al., 2017*). Moreover, eIF3h and specifically modified eIF3a have been directly shown to modulate resumption of scanning from *ATF4*'s uORF1 for efficient reinitiation downstream; that is one of the key steps in the *ATF4* translational control (*Hronová et al., 2017*; *Shu et al., 2022*). Therefore, we propose that the intact eIF3 complex is required for stimulation of reinitiation on uORF-containing mRNAs, such as *ATF4* and *MDM2* to ensure their efficient translation on demand. Interestingly, the impact on translation of *MDM2* mRNA could indicate that eIF3 is also linked to the key tumor suppressor p53 (*Marine and Lozano, 2010*).

Among a few upregulated DTEGs in eIF3h$^{KD}$, we noticed PRRC2A, which was significantly upregulated in all three knock-downs and was recently identified by the Teleman's group as a protein interacting with the translational machinery, including eIF3. Interestingly, they found that PRRC2 proteins affected translation initiation by promoting leaky scanning (*Bohlen et al., 2023*) and that the triple knock-down of all PRRC2 proteins (PRRC2A, PRRC2B, PRRC2C) specifically decreased translation efficiency of mRNAs harboring uORFs in their 5' UTRs, including *ATF4* and *RAF1* studied also here, similar to our eIF3 knock-downs. They concluded that uORFs are a key element determining PRRC2-dependence. Thus, we speculate that by increased expression of PRRC2A (and also PRRC2B and PRRC2C in eIF3d$^{KD}$ and eIF3e$^{KD}$), cells attempt to compensate for ineffective translation of mRNAs with short uORFs, which often encode regulatory proteins. Intriguingly, all PRRC2 proteins themselves have a highly translated overlapping-uORF in their 5' UTR, suggesting that they might be a subject to an autoregulatory mechanism.

Although our results presented here and summarized in *Figure 7F* are limited to HeLa cells and future work is needed to confirm their universal validity in other cells and tissues, a growing body of evidence suggests that eIF3 plays a critical role in translational regulation of specific mRNAs encoding regulatory proteins, very often utilizing uORFs and/or specific secondary structures in their 5' UTRs. Here, we demonstrate that alterations in the eIF3 subunit stoichiometry and/or eIF3 subcomplexes have distinct effects on the translatome; for example, they affect factors that play a prominent (either positive or negative) role in cancer biology (e.g. MDM2 and cJUN), but the resulting impact is unclear so far. Considering the complex interactions between these factors as well as the complexity of the eIF3 complex per se, future studies are required to delineate the specific oncogenic and tumor suppressive pathways that play a predominant role in mediating the effects of perturbations in the eIF3 complex

in the context of neoplasia. Taking into account that malignant cells exhibit augmented activity of many components of the translation machinery and are inherently believed to become 'addicted' to elevated protein synthesis (*Ruggero, 2013*; *Silvera et al., 2010*), this type of research is important because it is increasingly apparent that targeting general components of the canonical translational machinery, such as eIF3, which acts upstream of effector proteins directly involved in tumorigenesis, may hold promise for overcoming the major obstacle associated with intratumor heterogeneity.

# Materials and methods

## Cell lines, culture conditions, and transfection

HeLa cells (ATCC Cat# CCL-2, RRID:CVCL_0030) were grown at 37 °C and 5% $CO_2$ in Ø 15 cm dishes in DMEM (Sigma, cat #D6429) supplemented with 10% FBS (Gibco, cat # 10270–106). Twenty-four hours after seeding, cells were transfected with the ON-TARGETplus siRNA cocktail system from Dharmacon at a final concentration of 5 nM. Catalog numbers for all siRNAs used in this study are listed in *Supplementary file 3A*. INTERFERin (Polyplus, cat # 101000016) was used as a transfection reagent, 100 µl per dish. Cells were harvested 72 hr after transfection at approximately 80% confluency and cytoplasmatic lysates were prepared as described previously (*Herrmannová et al., 2020*). For monitoring levels of phosphorylated c-Jun protein, cytoplasmatic and nuclear lysates were prepared using NE-PER Nuclear and Cytoplasmic Extraction Reagents from Thermo Scientific.

## Ribosome profiling (Ribo-Seq) and RNA-seq library preparation

For *footprint (FP) libraries* cycloheximide was added to a final concentration of 100 µg/ml 1 min prior to harvesting. Cells were washed with ice cold 1 x PBS and lysed in buffer A (10 mM HEPES [pH 7.5], 62.5 mM KCl, 2.5 mM $MgCl_2$, 1 mM DTT, 1 mM PMSF, 1 µg/ml Aprotinin, 1 µg/ml Leupeptin, 1 µg/ml Pepstatin, Complete Mini EDTA-free [Roche, cat # 11836170001] – 1 tablet/5 ml, 1% Triton X-100, 100 µg/ml Cycloheximide) on the dish. The resulting lysate was collected and cleared by centrifugation. Aliquots of twelve absorbance units (AU) $OD_{260}$ of WCE were flash frozen in liquid nitrogen and used for control polysome profile and footprint isolation. An aliquot of 1 mg total protein was taken for control western blot, and another 200 µl aliquot was mixed with 1 ml of RNA Blue (TopBio, cat # R013) and taken for RNA isolation and control qPCR. When the lysate passed all three control experiments (polysome profile, western blot, qPCR), one aliquot of 12 AU $OD_{260}$ was thawed on ice, digested with 5 µl RNaseI (Ambion, cat # AM2295) for 30 min at 24 °C with mild shaking (300 rpm) and inactivated with SuperaseIN (Ambion, cat #AM2696). 80 S ribosomes were separated by high-velocity sedimentation through a 5% to 45% sucrose gradient at 39,000 rpm for 2.5 hr using the SW41Ti rotor. The gradients were scanned at 254 nm to visualize the ribosomal species and fractions containing 80 S ribosomes were collected and mixed with RNA Blue. RNA was isolated according to vendor's instructions, FP RNA was size selected on 15% TBE-Urea PAGE (Biorad, cat # 3540092) in range 20–40 nt and footprint library was prepared according to *Ingolia et al., 2012*. FP libraries were sequenced on NovaSeq6000 (Illumina) with read length 50 nt.

For *mRNA libraries* the growth media was aspirated and cells were harvested into 5 ml of RNA Blue per Ø 15 cm dish. RNA was isolated according to vendor's instructions. mRNA was isolated from the total RNA using Poly(A)purist MAG kit (Ambion, cat # AM1922), fragmented using RNA Fragmentation Reagents (Ambion, cat # AM8740) and size selected on 15% TBE-Urea PAGE in range 50–100 nt. Size selected mRNA was then dephosphorylated and library was prepared using SMARTer smRNA-Seq Kit (TAKARA, cat # 635029) according to vendor's instructions. mRNA libraries were sequenced on NovaSeq6000 (Illumina) with read length 150 nt.

## RNA isolation, reverse transcription, and qPCR

Total RNA was isolated using RNA Blue reagent (TopBio, cat # R013) 72 hr post-transfection according to the manufacturer's instructions. After DNase I digestion (NEB, cat # M0303L), cDNA was synthesized from 1 µg RNA using the High-Capacity cDNA Reverse Transcription Kit (Applied Biosystems, # 4368813). qPCR was performed in triplicates using 5×HOT FIREPol EvaGreen qPCR Mix Plus (Solis BioDyne # 08-25-00020). The obtained data were normalized to ALAS1 mRNA levels and non-transfected control. All qPCR primers are listed in *Supplementary file 3B*.

## Western blotting

All samples were resolved using SDS-PAGE followed by western blotting. All primary antibodies used in this study are listed in *Supplementary file 3C*. The signal was developed using SuperSignal West Femto Maximum Sensitivity Substrate (Thermo Fisher Scientific, cat # 34096) and detected in a G-Box imager (Syngene) using a series of varying exposure times. Signals were processed with Quantity One (Bio-Rad). Only signals from the same strips and with the same exposure times were compared.

## Polysome profiles

One aliquot of 12 AU $OD_{260}$ was thawed on ice and separated by high-velocity sedimentation through a 5% to 45% sucrose gradient at 39,000 rpm for 2.5 hr using the SW41Ti rotor. The gradients were scanned at 254 nm to visualize the ribosomal species.

## 60S/40S ratio calculation

Cell lysates were prepared as above by on dish lysis in buffer A. To 6 AU $OD_{260}$ EDTA was added to final concentration of 50 mM and the lysate was separated by high-velocity sedimentation through a 5% to 50% sucrose gradient at 39,000 rpm for 3 h using the SW41Ti rotor. The gradients were scanned at 260 nm to visualize the ribosomal species. Area under the curve was quantified to calculate the 60 S:40 S ratio to check for ribosome biogenesis defect.

## Ribosome content profiles

Cells were counted by Corning Automated Cell Counter and 10 million cells were pelleted in microcentrifuge tube and lysed in buffer A (as above). EDTA was added to the lysate to final concentration of 50 mM and the lysate was separated by high-velocity sedimentation through a 5% to 50% sucrose gradient at 39,000 rpm for 3.5 hr using the SW41Ti rotor. The gradients were scanned at 260 nm to visualize the ribosomal species. Area under the curve was quantified to calculate the total amount of ribosomes between samples.

## Analysis of sequencing data

We processed the ribosome *footprint libraries* as follows. Adapter sequence 'CTGTAGGCACCATCAA T' and all sequences 3' of the adapter were trimmed from the 3' end of each read by Cutadapt v. 3.5 (*Martin, 2011*). Flanking Ns were trimmed from both 5' and 3' ends of each read. Reads without adapter sequence and reads shorter than 15 nt after trimming were discarded. Next, rRNA and tRNA reads were discarded after alignment with Bowtie2 v. 2.4.4 (*Langmead and Salzberg, 2012*). For rRNA, we used RNA5S1, RNA5-8SN5, RNA18SN5 and RNA28SN5 from NCBI (*Sayers et al., 2022*), and ENSG00000211459 and ENSG00000210082 from Ensembl (*Howe et al., 2021*). For tRNA we used all human tRNA sequences from tRNAdb (*Jühling et al., 2009*) and all high confidence hg38 tRNA sequences from GtRNAdb (*Chan and Lowe, 2016*). Remaining reads were aligned to genome and transcriptome from Ensembl (GRCh38.p13, annotation release 104) by the STAR aligner v. 2.7.9 a_2021-06-25 (*Dobin et al., 2013*) using local alignment with 10% mismatches, and with indels and soft clipping allowed for transcriptomic alignment. Gene counts were counted from genomic alignment using Samtools index v. 1.13 (*Danecek et al., 2021*) and Htseq-count v. 0.13.5 (*Anders et al., 2015*) within CDS regions. Transcriptomic alignments were further processed using MANE Select set from MANE project v. 0.95 (*Morales et al., 2022*) and custom programs 'mane2ensembl_ gtf', 'filter_reverse_reads', 'filter_ambiguous_genes' and 'select_transcripts'. See *Supplementary file 4* for detailed overview of raw read processing.

The *RNA-seq libraries* were processed the same as above, except that raw read processing by Cutadapt had to be adapted to different library construction method (SMARTer smRNA-Seq Kit, TAKARA). In particular, first 3 nucleotides on the 5' end were removed, and poly(A) sequence (10xA) was trimmed from the 3'end.

Quality checks were done by FastQC v. 0.11.9 (*Andrews, 2010*) and Ribo-seQC v. 1.1 (*Calviello et al., 2019*). Triplet periodicity was checked on transcriptomic alignments using custom modified riboWaltz (*Lauria et al., 2018*) and a custom script 'Triplet_periodicity.R'. Gene counts were used to check correlation between replicates using a custom script 'Correlation-samples.R'. Read length counts were used to check read length distribution using a custom script 'Lengths.R'.

PCA plot, sample distances clustering and the differential expression analysis were performed on gene counts by DESeq2 (*Love et al., 2014*) with biomaRt library (*Durinck et al., 2009*) used to download gene annotations. Differential translation efficiency (TE) values between NT and the knock-downs were calculated as described in *Chothani et al., 2019*. The $p_{adj}$ threshold for differentially translated transcripts was set to 0.05. For details see custom script 'Differential analysis.R'. Results of differential expression analysis were further used to generate heatmap and scatterplots of differentially expressed genes and genes with defined p-adjusted value were used to analyze dependency between mRNA characteristics and changes in TE using custom scripts 'Significant_TE_heatmap.R', 'Significant_TE_intersection.R' and 'Correlation-features.R'. To get mRNA characteristics we used MANE Select set to select a single representative transcript for each protein-coding gene. uORF database uORFdb (*Manske et al., 2023*) was used to get number of ATG uORFs per mRNA.

Metagene analysis was done as described in *Lin et al., 2020*, but 5' ends of reads were used instead of P-sites. Samtools and custom programs 'transcripts_startstop_positions', 'read_counts' 'nd 'region_readcounts' were used to get read counts relative to the start codon. These were normalized to a number of reads in a region starting 75 bases after the start codon and ending 15 bases before the stop codon and averaged over the samples. Only transcripts that met the following parameters were considered: CDS longer than 450 nt and RPKM higher than 10. Weighted counts of biological replicates were further used to compute average read counts for each sample. Moving average method was used with a 21 bases window size.

For footprint coverage plots of individual mRNAs, MANE transcripts of examined genes (ENST00000674920-ATF4; ENST00000258149-MDM2; ENST00000251849-RAF1) were selected. Reads were counted for each position using Samtools and a custom script read_counts and normalized to number of read counts over all positions of transcript. Weighted counts of biological replicates were further used to compute average read counts for each sample. Graphs were smoothened with a sliding window of 30nt. All used custom scripts are available on GitHub, copy archived at *Jelinek, 2024*.

## KEGG/GO pathway enrichment analysis

We analyzed enrichment of the differentially translated transcripts in KEGG pathways (*Kanehisa and Sato, 2020*) and Gene Ontology (*Ashburner et al., 2000*; *Carbon et al., 2021*), using Enrichr (*Kuleshov et al., 2016*), ShinyGO v0.741 (*Ge et al., 2020*) and KEGG Mapper Search (*Kanehisa et al., 2022*). Venn diagrams were created using InteractiVenn (*Heberle et al., 2015*).

## Analysis of gene signatures

To assess changes in the regulation of known gene signatures in the eIF3 KDs, empirical cumulative distribution functions (ECDFs) of $\log_2$ fold changes in FP and total mRNA were plotted independently. Fold change ECDFs for genes belonging to a particular signature were compared to those for all other genes, and differences in distributions were calculated at the quantiles. Significant directional shifts between the signatures and the background were identified using the Wilcoxon rank-sum test.

## Acknowledgements

We are thankful to Olga Krýdová for technical and administrative assistance and to all lab members for fruitful discussions. We are also thankful to Jonathan Bohlen for sharing and discussing results of PRRC2 proteins downregulation. This work was supported by a Grant of Excellence in Basic Research (EXPRO 2019) provided by the Czech Science Foundation (19-25821X), the Praemium Academiae grant provided by the Czech Academy of Sciences, and CZ.02.01.01/00/22_008/0004575 RNA for therapy by ERDF and MEYS (all to LSV), Czech Science Foundation grant 19-08013S (to TV), ELIXIR CZ research infrastructure project (MEYS Grant No: LM2023055) including access to computing and storage facilities, by National Institute for Cancer Research (Programme EXCELES, ID Project No. LX22NPO5102) funded by the European Union - Next Generation EU (to JB) and by Fonds de Recherche du Québec – Santé (FRQS) Senior Investigator award (awarded to IT).

## Additional information

### Competing interests
Ivan Topisirovic: Reviewing editor, *eLife*. The other authors declare that no competing interests exist.

### Funding

| Funder | Grant reference number | Author |
| --- | --- | --- |
| Grantová Agentura České Republiky | 19-25821X | Leoš Shivaya Valášek |
| Akademie Věd České Republiky | Praemium Academiae | Leoš Shivaya Valášek |
| Ministerstvo Školství, Mládeže a Tělovýchovy | CZ.02.01.01/00/22_008/0004575 | Leoš Shivaya Valášek |
| Grantová Agentura České Republiky | 19-08013S | Tomáš Vomastek |
| Fonds de Recherche du Québec - Santé | Senior Investigator award | Ivan Topisirovic |
| The Ministry of Education, Youth and Sports | LM2023055 | Jan Jelínek |
| European Union | LX22NPO5102 | Jan Brábek |

The funders had no role in study design, data collection and interpretation, or the decision to submit the work for publication.

### Author contributions
Anna Herrmannová, Conceptualization, Formal analysis, Validation, Investigation, Methodology, Writing – original draft, Writing – review and editing; Jan Jelínek, Data curation, Software, Formal analysis, Validation, Investigation, Methodology, Writing – original draft; Klára Pospíšilová, Mahabub Pasha Mohammad, Formal analysis, Investigation; Farkas Kerényi, Tomáš Vomastek, Kathleen Watt, Formal analysis, Investigation, Methodology; Jan Brábek, Formal analysis; Susan Wagner, Formal analysis, Methodology; Ivan Topisirovic, Formal analysis, Validation; Leoš Shivaya Valášek, Conceptualization, Formal analysis, Supervision, Funding acquisition, Validation, Writing – original draft, Project administration, Writing – review and editing

### Author ORCIDs
Anna Herrmannová https://orcid.org/0000-0003-3500-3212
Kathleen Watt https://orcid.org/0000-0001-6642-9756
Ivan Topisirovic https://orcid.org/0000-0002-5510-9762
Leoš Shivaya Valášek https://orcid.org/0000-0001-8123-8667

Reviewer #1 (Public review): https://doi.org/10.7554/eLife.95846.3.sa1
Reviewer #2 (Public review): https://doi.org/10.7554/eLife.95846.3.sa2
Reviewer #3 (Public review): https://doi.org/10.7554/eLife.95846.3.sa3
Author response https://doi.org/10.7554/eLife.95846.3.sa4

## Additional files

### Supplementary files
• Supplementary file 1. Complete differential expression data.
• Supplementary file 2. KEGG pathway and GO Biological Process enrichment analysis of eIF3h$^{KD}$ downregulated DTEGs.
• Supplementary file 3. Material tables – siRNAs, qPCR primers and antibodies used in this study.
• Supplementary file 4. Statistics of read processing for all Ribo-Seq and RNA-Seq libraries.

• MDAR checklist

## Data availability

The datasets generated during this study were deposited in the GEO database with the following accession number: GSE216967. All custom programs source code are published at GitHub, copy archived at *Jelinek, 2024*.

The following dataset was generated:

| Author(s) | Year | Dataset title | Dataset URL | Database and Identifier |
|---|---|---|---|---|
| Herrmannová A, Jelínek J, Kerényi F, Pospíšilová K, Wagner S, Brábek J, Valášek LS | 2024 | Differential expression analysis connects knock-downs of eIF3e and eIF3d with the MAP kinase signaling pathway | http://www.ncbi.nlm.nih.gov/geo/query/acc.cgi?acc=GSE216967 | NCBI Gene Expression Omnibus, GSE216967 |

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
