## [Editor Report · eLife Assessment]

This study demonstrates mRNA-specific regulation of translation by subunits of the eukaryotic initiation factor complex 3 (eIF3) using **convincing** methods, data, and analyses. The investigations have generated **important** information that will be of interest to biologists studying translation regulation. However, the physiological significance of the gene expression changes that were observed is not clear.

---

## [Referee Report · Reviewer #1 (Public review)]

Summary:

In this manuscript, Herrmannova et al explore changes in translation upon individual depletion of three subunits of the eIF3 complex (d, e and h) in mammalian cells. The authors provide a detailed analysis of regulated transcripts, followed by validation by RT-qPCR and/or Western blot of targets of interest, as well as GO and KKEG pathway analysis. The authors confirm prior observations that eIF3, despite being a general translation initiation factor, functions in mRNA-specific regulation, and that eIF3 is important for translation re-initiation. They show that global effects of eIF3e and eIF3d depletion on translation and cell growth are concordant. Their results support and extend previous reports suggesting that both factors control translation of 5'TOP mRNAs. Interestingly, they identify MAPK pathway components as a group of targets coordinately regulated by eIF3 d/e. The authors also discuss discrepancies with other reports analyzing eIF3e function.

Strengths:

Altogether, a solid analysis of eIF3 d/e/h-mediated translation regulation of specific transcripts. The data will be useful for scientists working in the Translation field.

Weaknesses:

The authors could have explored in more detail some of their novel observations, as well as their impact on cell behavior.

The manuscript has improved with the new corrections. I appreciate the authors' attention to the minor comments, which have been fully solved. The authors have not, however, provided additional experimental evidence that uORF-mediated translation of Raf-1 mRNA depends on an intact eIF3 complex, nor have they addressed the consequences of such regulation for cell physiology. While I understand that this is a subject of follow-up research, the authors could have at least included their explanations/ speculations regarding major comments 2-4, which in my opinion could have been useful for the reader.

---

## [Referee Report · Reviewer #2 (Public review)]

Summary:

mRNA translation regulation permits cells to rapidly adapt to diverse stimuli by fine tuning gene expression. Specifically, the 13-subunit eukaryotic initiation factor 3 (eIF3) complex is critical for translation initiation as it aids in 48S PIC assembly to allow for ribosome scanning. In addition, eIF3 has been shown to drive transcript-specific translation by binding mRNA 5' cap structures through the eIF3d subunit. Dysregulation of eIF3 has been implicated in oncogenesis, however the precise eIF3 subunit contributions are unclear. Here, Herrmannová et al. aim to investigate how eIF3 subcomplexes, generated by knock down (KD) of either eIF3e, eIF3d or eIF3h, affect the global translatome. Using Ribo-seq and RNA-seq, the authors identified a large number of genes that exhibit altered translation efficiency upon eIF3d/e KD, while translation defects upon eIF3h KD were mild. eIF3d/eKD share multiple dysregulated transcripts, perhaps due to both subcomplexes lacking eIF3d. Both eIF3d/e KD increase translation efficiency (TE) of transcripts encoding lysosomal, ER and ribosomal proteins, suggesting a role of eIF3 in ribosome biogenesis and protein quality control. Many transcripts encoding ribosomal proteins harbor a TOP motif, and eIF3d KD and eIF3e KD cells exhibit a striking induction of these TOP-modified transcripts. On the other hand, eIF3d KD and eIF3e KD leads to a reduction of MAPK/ERK pathway proteins. Despite this downregulation, eIF3d KD and eIF3e KD activates MAPK/ERK signaling as ERK1/2 and c-Jun phosphorylation was induced. Finally, in all three knockdowns, MDM2 and ATF4 protein levels are reduced. This is notable because MDM2 and ATF4 both contain short uORFs upstream of the start codon, and further supports a role of eIF3 in reinitiation. Altogether, Herrmannová et al. have gained key insights to precise eIF3-mediated translational control as it relates to key signaling pathways implicated in cancer.

Strengths:

The authors have provided a comprehensive set of data to analyze RNA and ribosome footprinting upon perturbation of eIF3d, eIF3e, and eIF3h. As described above in the summary, these data present many interesting starting points to understand additional roles of the eIF3 complex and specific subunits in translational control.

Weaknesses:

- The differences between eIF3e and eIF3d knockdown are difficult to reconcile, especially since eIF3e knockdown leads to reduction in eIF3d levels.

- The paper would be strengthened by experiments directly testing what RNA determinants allow for transcript-specific translation regulation by the eIF3 complex. This would allow the paper to be less descriptive.

- The paper would have more biological relevance if eIF3 subunits were perturbed to mimic naturally occurring situations where eIF3 is dysregulated. For example, eIF3e is aberrantly upregulated in certain cancers, and therefore an overexpression and profiling experiment would have been more relevant than a knockdown experiment.

The first review is unchanged as no additional experiments were provided to address the first review.

---

## [Referee Report · Reviewer #3 (Public review)]

Summary:

In this article, Hermannova et al catalog the changes in ribosome association with mRNAs when the multisubunit eukaryotic translation initiation factor 3 is disrupted by knocking down individual subunits. They find that RNAs relying on TOP motifs for translation, such as ribosomal protein RNAs, and RNAs encoding modification enzymes in the ER and components of the lysosome are upregulated. In contrast, proteins encoding components of MAP kinase cascades are downregulated when subunits of eIF3 are knocked down, but retain elevated levels of activity.

Strengths:

The authors use ribosome profiling of well-characterized mutants lacking subunits of eIF3 and assess the changes in translation that take place. They supplement the ribosome association studies with western blotting to determine protein level changes of affected transcripts. They analyze what transcripts undergo translation changes, which is important for understanding more broadly how translation initiation factor levels affect cancer cell translatomes. Changes observed by both ribosome profiling and western blotting supports their claims that eIF3 functions in mRNA-specific control of translation.

Weaknesses:

(1) The paper would be strengthened if there were a clear model tying the various effects together or linking individual subunit knockdown to cancerous phenotypes. It is noted that the authors plan to address such outcomes of eIF3 dysregulation in future work, which will be of interest.

(2) The paper could also be strengthened if some of the experiments were performed in at least one other cell type to determine whether changes observed are general or cell-type specific. The authors discuss this issue and provide a literature citation to support a more general mechanism.

---

## [Author Response]

The following is the authors’ response to the current reviews.

**Reviewer #1 (Recommendations for the authors):**
The manuscript has improved with the new corrections. I appreciate the authors' attention to the minor comments, which have been fully solved. The authors have not, however, provided additional experimental evidence that uORF-mediated translation of Raf-1 mRNA depends on an intact eIF3 complex, nor have they addressed the consequences of such regulation for cell physiology. While I understand that this is a subject of follow-up research, the authors could have at least included their explanations/ speculations regarding major comments 2-4, which in my opinion could have been useful for the reader.

Our explanations/speculations regarding major comments 2 and 3 were included in the Discussion. We apologize for this misunderstanding as we thought that we were supposed to explain our ideas only in the responses. We did not discuss the comment 4, however, as we are really not sure what is the true effect and did not want to go into wild speculations in our manuscript. We thank this reviewer for his insightful comments and understanding.

The following is the authors’ response to the original reviews.

**Reviewer #1 (Recommendations For The Authors):**
Major comments:(1) The authors report the potential translational regulation of Raf kinase by re-initiation. It would be interesting to show that Raf is indeed regulated by uORF-mediated translation, and that this is dependent on an intact eIF3 complex. Analyzing the potential consequences of Raf1 regulation for cancer cell proliferation or apoptosis would be a plus.

We agree that this is an interesting and likely possibility. In fact, another clue that translation of Raf1 is regulated by uORFs comes from Bohlen et al. 2023 (PMID: 36869665) where they showed that RAF1 translation is dependent on PRRC2 proteins (that promote leaky scanning through these uORFs). We noted in the discussion that our results from eIF3d/e/hKD and the PRRC2A/B/CKD partly overlap. It is a subject of our follow-up research to investigate whether eIF3 and PRRC2 co-operate together to regulate translation of this important mRNA.

(2) The authors show that eIF3 d/e -but not 3h- has an effect on cell proliferation. First, this indicates that proliferation does not fully correlate with eIF3 integrity. Depletion of eIF3d does not affect the integrity of eIF3, yet the effects on proliferation are similar to those of eIF3e. What is the possibility that changes in proliferation reflect functions of eIF3d outside the eIF3 complex? What could be the real consequences of disturbing eIF3 integrity for the mammalian cell? Please, discuss.

Yes, proliferation does not fully correlate with eIF3 integrity. Downregulation of eIF3 subunits that lead to disintegration of eIF3 YLC core (a, b, c, g, i) have more detrimental effect on growth and translation than downregulation of the peripheral subunits (e, k, l, f, h, m). Our previous studies (Wagner et al. 2016, PMID: 27924037 and Herrmannová et al. 2020, PMID: 31863585) indicate that the YLC core of eIF3 can partially support translation even without its peripheral subunits. In this respect eIF3d (as a peripheral subunit) is an amazing exception, suggesting it may have some specialized function(s). Whether this function resides outside of the eIF3 complex or not we do not know, but do not think so. Mainly because in the absence of eIF3e – its interaction partner, eIF3d gets rapidly degraded. Therefore, it is not very likely that eIF3d exists alone outside of eIF3 complex with moonlighting functions elsewhere. We think that eIF3d, as a head-interacting subunit close to an important head ribosomal protein RACK1 (a landing pad for regulatory proteins), is a target of signaling pathways, which may make it important for translation of specific mRNAs. In support is these thoughts, eIF3d (in the context of entire eIF3) together with DAP5 were shown to promote translation by an alternate capdependent (eIF4F-independent) mechanism (Lee et al. 2016, PMID: 27462815; de la Parra et al. 2018, PMID:30076308). In addition, the eIF3d function (also in the context of entire eIF3) was proved to be regulated by stress-triggered phosphorylation (Lamper et al. 2020, PMID: 33184215).

(3) Figure 6D: Surprisingly, reduced levels of ERK1/2 upon eIF3d/e-KD are compensated by increased phosphorylation of ERK1/2 and net activation of c-Jun. Please comment on the functional consequences of buffering mechanisms that the cell deploys in order to counteract compromised eIF3 function. Why would the cell activate precisely the MAPK pathway to compensate for a compromised eIF3 function?

This we do not know. We can only speculate that when translation is compromised, cells try to counteract it in two ways: (1) they produce more ribosomes to increase translational rates and (2) activate MAPK signaling to send pro-growth signals, which can in the end further boost ribosome biogenesis.

(4) Regarding DAP-sensitive transcripts, can the authors discuss in more detail the role of eIF3d in alternative cap-dependent translation versus re-initiation? Are these transcripts being translated by a canonical cap- and uORF-dependent mechanism or by an alternative capdependent mechanism?

This is indeed not an easy question. On one hand, it was shown that DAP5 facilitates translation re-initiation after uORF translation in a canonical cap-dependent manner. This mechanism is essential for translation of the main coding sequence (CDS) in mRNAs with structured 5' leaders and multiple uORFs. (Weber et al. 2022, PMID: 36473845; David et al., 2022, PMID: 35961752). On the other hand, DAP5 was proposed to promote alternative, eIF4F-independent but cap-dependent translation, as it can substitute the function of the eIF4F complex in cooperation with eIF3d (de la Parra et al., 2018, PMID: 30076308; Volta et al., 2021 34848685). Overall, these observations paint a very complex picture for us to propose a clear scenario of what is going on between these two proteins on individual mRNAs. We speculate that both mechanisms are taking place and that the specific mechanism of translation initiation differs for differently arranged mRNAs.

Minor comments:(5) Figure S2C: why is there a strong reduction of the stop codon peak for 3d and 3h KDs?

We have checked the Ribowaltz profiles of all replicates (in the Supplementary data we are showing only a representative replicate I) and the stop codon peak differs a lot among the replicates. We think that this way of plotting was optimized for calculation and visualization of P-sites and triplet periodicity and thus is not suitable for this type of comparison among samples. Therefore, we have performed our own analysis where the 5’ ends of reads are used instead of P-sites and triplicates are averaged and normalized to CDS (see below please), so that all samples can be compared directly in one plot (same as Fig. S13A but for stop codon). We can see that the stop codon peak really differs and is the smallest for eIF3hKD. However, these changes are in the range of 20% and we are not sure about their biological significance. We therefore refrain from drawing any conclusions. In general, reduced stop codon peak may signal faster termination or increased stop codon readthrough, but the latter should be accompanied by an increased ribosome density in the 3’UTR, which is not the case. A defect in termination efficiency would be manifested by an increased stop codon peak, instead.

**Author response image 1. sa4fig1:** 

(6) Figures 5 and S8: Adding a vertical line at 'zero' in all cumulative plots will help the reader understand the author's interpretation of the data.

We have added a dashed grey vertical line at zero as requested. However, for interpretation of these plots, the reader should focus on the colored curve and whether it is shifted in respect to the grey curve (background) or not. Shift to the right indicates increased expression, while shift to the left indicates decreased expression. The reported p-value then indicates the statistical significance of the shift.

(7) The entire Figure 2 are controls that can go to Supplementary Material. The clustering of Figure S3B could be shown in the main Figure, as it is a very easy read-out of the consistent effects of the KDs of the different eIF3 subunits under analysis.

We have moved the entire Figure 2 to Supplementary Material as suggested (the original panels can be found as Supplementary Figures 1B, 1C and 3A). Figure S3B is now the main Figure 2E.

(8) There are 3 replicates for Ribo-Seq and four for RNA-Seq. Were these not carried out in parallel, as it is usually done in Ribo-seq experiments? Why is there an extra replicate for RNASeq?

Yes, the three replicates were carried out in parallel. We have decided to add the fourth replicate in RNA-Seq to increase the data robustness as the RNA-Seq is used for normalization of FP to calculate the TE, which was our main analyzed metrics in this article. We had the option to add the fourth replicate as we originally prepared five biological replicates for all samples, but after performing the control experiments, we selected only the 3 best replicates for the Ribo-Seq library preparation and sequencing.

(9) Please, add another sheet in Table S2 with the names of all genes that change only at the translation (RPF) levels.

As requested, we have added three extra sheets (one for each downregulation) for differential FP with Padjusted <0.05 in the Spreadsheet S2. We also provide a complete unfiltered differential expression data (sheet named “all data”), so that readers can filter out any relevant data based on their interest.

(10) Page 5, bottom: ' ...we showed that the expression of all 12 eIF3 subunits is interconnected such that perturbance of the expression of one subunit results in the down-regulation of entire modules...'. This is not true for eIF3d, as shown in Fig1B and mentioned in Results.

This reviewer is correct. By this generalized statement, we were trying to summarize our previous results from Wagner et al., 2014, PMID: 24912683; Wagner et al.,2016, PMID: 27924037 and Herrmannova et al.,2020, PMID: 31863585. The eIF3d downregulation is the only exception that does not affect expression of any other eIF3 subunit. Therefore, we have rewritten this paragraph accordingly: “We recently reported a comprehensive in vivo analysis of the modular dynamics of the human eIF3 complex (Wagner *et al*, 2020; Wagner *et al*, 2014; Wagner *et al.*, 2016). Using a systematic individual downregulation strategy, we showed that the expression of all 12 eIF3 subunits is interconnected such that perturbance of the expression of one subunit results in the down-regulation of entire modules leading to the formation of partial eIF3 subcomplexes with limited functionality (Herrmannova *et al*, 2020). eIF3d is the only exception in this respect, as its downregulation does not influence expression of any other eIF3 subunit.”

(11) Page 10, bottom: ' The PCA plot and hierarchical clustering... These results suggest that eIF3h depletion impacts the translatome differentially than depletion of eIF3e or eIF3d.' This is already obvious in the polysome profiles of Figure S2C.

We agree that this result is surely not surprising given the polysome profile and growth phenotype analyses of eIF3hKD. But still, we think that the PCA plot and hierarchical clustering results represent valuable controls. Nonetheless, we rephrased this section to note that this result agrees with the polysome profiles analysis: “The PCA plot and hierarchical clustering (Figure 2A and Supplementary Figure 4A) showed clustering of the samples into two main groups: Ribo-Seq and RNA-seq, and also into two subgroups; NT and eIF3hKD samples clustered on one side and eIF3eKD and eIF3dKD samples on the other. These results suggest that the eIF3h depletion has a much milder impact on the translatome than depletion of eIF3e or eIF3d, which agrees with the growth phenotype and polysome profile analyses (Supplementary Figure 1A and 1D).”

(12) Page 12: ' As for the eIF3dKD "unique upregulated" DTEGs, we identified one interesting and unique KEGG pathway, the ABC transporters (Supplementary Figure 5A, in green).' This sentence is confusing, as there are more pathways that are significant in this group, so it is unclear why the authors consider it 'unique'.

The eIF3dKD “unique upregulated” group comprises genes with increased TE only in eIF3dKD but not in eIF3eKD or eIF3hKD (500 genes, Fig 2G). All these 500 genes were examined for enrichment in the KEGG pathways, and the top 10 significant pathways were reported (Fig S6A). However, 8 out of these 10 pathways were also significantly enriched in other gene groups examined (e.g. eIF3d/eIF3e common). Therefore, the two remaining pathways (“ABC transporters” and “Other types of O-glycan biosynthesis”) are truly unique for eIF3dKD. We wanted to highlight the ABC transporters group in particular because we find it rather interesting (for the reasons mentioned in the article). We have corrected the sentence in question to avoid confusion: “Among the eIF3dKD “unique upregulated” DTEGs, we identified one interesting KEGG pathway, the ABC transporters, which did not show up in other gene groups (Supplementary Figure 6A, in green). A total of 12 different ABC transporters had elevated TE (9 of them are unique to eIF3dKD, while 3 were also found in eIF3eKD), 6 of which (ABCC1-5, ABCC10) belong to the C subfamily, known to confer multidrug resistance with alternative designation as multidrug resistance protein (MRP1-5, MRP7) (Sodani *et al*, 2012).

Interestingly, all six of these ABCC transporters were upregulated solely at the translational level (Supplementary Spreadsheet S2).”

(13) Note typo ('Various') in Figure 4A.

Corrected

(14) The introduction could be shortened.

This is a very subjective requirement. In fact, when this manuscript was reviewed in NAR, we were asked by two reviewers to expand it substantially. Because a number of various research topics come together in this work, e.g. translational regulation, the eIF3 structure and function, MAPK/ERK signaling, we are convinced that all of them demand a comprehensive introduction for non-experts in each of these topics. Therefore, with all due respect to this reviewer, we did not ultimately shorten it.

**Reviewer #2 (Recommendations For The Authors):**
- In Figure 2, it would be useful to know why eIF3d is destabilized by eIF3e knockdown - is it protein degradation and why do the eIF3d/e knockdowns not more completely phenocopy each other when there is the same reduction to eIF3d as in the eIF3d knockdown sample?

Yes, we do think that protein degradation lies behind the eIF3d destabilization in the eIF3eKD, but we have not yet directly demonstrated this. However, we have shown that eIF3d mRNA levels are not altered in eIF3eKD and that Ribo-Seq data indicate no change in TE or FP for eIF3d-encoding mRNA in eIF3eKD. Nonetheless, it is important to note (and we discuss it in the article) that eIF3d levels in eIF3dKD are lower than eIF3d levels in eIF3eKD (please see Supplementary Figure 1C). In fact, we believe that this is one of the main reasons for the eIF3d/e knockdowns differences.

- The western blots in Figures 4 and 6 show modest changes to target protein levels and would be strengthened by quantification.

We have added the quantifications as requested by this reviewer and the reviewer 3.

- For Figure 4, this figure would be strengthened by experiments showing if the increase in ribosomal protein levels is correlated with actual changes to ribosome biogenesis.

As suggested, we performed polysome profiling in the presence of EDTA to monitor changes in the 60S/40S ratio, indicating a potential imbalance in the biogenesis of individual ribosome subunits. We found that it was not affected (Figure 3G). In addition, we performed the same experiment, normalizing all samples to the same number of cells (cells were carefully counted before lysis). In this way, we confirmed that eIF3dKD and eIF3eKD cells indeed contain a significantly increased number of ribosomes, in agreement with the western blot analysis (Figure 3H).

- In Figure 6, there needs to be a nuclear loading control.

This experiment was repeated with Lamin B1 used as a nuclear loading control – it is now shown as Fig. 5F.

- For Figure 8, these findings would be strengthened using luciferase reporter assays where the various RNA determinants are experimentally tested. Similarly, 5′ TOP RNA reporters would have been appreciated in Figure 4.

This is indeed a logical continuation of our work, which represents the current work in progress of one of the PhD students. We apologize, but we consider this time- and resource-demanding analysis out of scope of this article.

**Reviewer #3 (Recommendations For The Authors):**
(1) Within the many effects observed, it is mentioned that eIF3d is known to be overexpressed while eIF3e is underexpressed in many cancers, but knockdown of either subunit decreases MDM2 levels, which would be expected to increase P53 activity and decrease tumor cell transformation. In contrast, they also report that 3e/3d knockdown dramatically increases levels of cJUN, presumably due to increased MAPK activity, and is expected to increase protumor gene expression. Additional discussion is needed to clarify the significance of the findings, which are a bit confusing.

This is indeed true. However, considering the complexity of eIF3, the largest initiation factor among all, as well as the broad portfolio of its functions, it is perhaps not so surprising that the observed effects are complex and may seem even contradictory in respect to cancer. To acknowledge that, we expanded the corresponding part of discussion as follows: “Here, we demonstrate that alterations in the eIF3 subunit stoichiometry and/or eIF3 subcomplexes have distinct effects on the translatome; for example, they affect factors that play a prominent (either positive or negative) role in cancer biology (e.g., MDM2 and cJUN), but the resulting impact is unclear so far. Considering the complex interactions between these factors as well as the complexity of the eIF3 complex *per se*, future studies are required to delineate the specific oncogenic and tumor suppressive pathways that play a predominant role in mediating the effects of perturbations in the eIF3 complex in the context of neoplasia.”

(2) There are places in the text where the authors refer to changes in transcriptional control when RNA levels differ, but transcription versus RNA turnover wasn't tested, e.g. page 16 and Figure S10, qPCR does not confirm "transcriptional upregulation in all three knockdowns" and page 19 "despite apparent compensatory mechanisms that increase their transcription."

This is indeed true, the sentences in question were corrected. The term “increased mRNA levels” was used instead of transcriptional upregulation (increased mRNA stabilization is also possible).

(3) Similarly, the authors suggest that steady-state LARP1 protein levels are unaffected based on ribosome footprint counts (page 21). It is incorrect to assume this, because ribosome footprints can be elevated due to stalling on RNA that isn't being translated and doesn't yield more protein, and because levels of translated RNA/synthesized proteins do not always reflect steady-state protein levels, especially in mutants that could affect lysosome levels and protein turnover. Also page 12, 1st paragraph suggests protein production is down when ribosome footprints are changed.

Yes, we are well-aware of this known limitation of Ribo-seq analysis. Therefore, the steadystate protein levels of our key hits were verified by western blotting. In addition, we have removed the sentence about LARP1 because it was based on Ribo-Seq data only without experimental evaluation of the steady-state LARP1 protein levels.

(4) The translation buffering effect is not clear in some Figures, e.g. S6, S8, 8A, and B. The authors show a scheme for translationally buffered RNAs being clustered in the upper right and lower left quadrants in S4H (translation up with transcript level down and v.v.), but in the FP versus RNA plots, the non-TOP RNAs and 4E-P-regulated RNAs don't show this behavior, and appear to show a similar distribution to the global changes. Some of the right panels in these figures show modest shifts, but it's not clear how these were determined to be significant. More information is needed to clarify, or a different presentation, such as displaying the RNA subsets in the left panels with heat map coloring to reveal whether RNAs show the buffered translation pattern defined in purple in Figure S4H, or by reporting a statistical parameter or number of RNAs that show behavior out of total for significance. Currently the conclusion that these RNAs are translationally buffered seems subjective since there are clearly many RNAs that don't show changes, or show translation-only or RNA-only changes.

We would like to clarify that S4H does not indicate a necessity for changes in FPs in the buffered subsets. Although opposing changes in total mRNA and FPs are classified as buffering, often we also consider the scenario where there are changes to the total mRNA levels not accompanied by changes in ribosome association.

In figure S6, the scatterplots indicate a high density of genes shifted towards negative fold changes on the x-axis (total mRNA). This is also reflected in the empirical cumulative distribution functions (ecdfs) for the log2 fold changes in total mRNA in the far right panels of A and B, and the lack of changes in log2 fold change for FPs (middle panels). Similarly, in figure S8, the scatterplots indicate a density of genes shifted towards positive fold changes on the x-axis for total mRNA. The ecdfs also demonstrate that there is a significant directional shift in log2 fold changes in the total mRNA that is not present to a similar degree in the FPs, consistent with translational offsetting. It is rightly pointed out that not all genes in these sets follow the same pattern of regulation. We have revised the title of Supplementary Figure S6 (now S7) to reflect this. However, we would like to emphasize that these figures are not intended to communicate that all genes within these sets of interest are regulated in the same manner, but rather that when considered as a whole, the predominant effect seen is that of translational offsetting (directional shifts in the log2 fold change distribution of total mRNA that are not accompanied by similar shifts in FP mRNA log2 fold changes).

The significance of these differences was determined by comparing the ecdfs of the log2 fold changes for the genes belonging to a particular set (e.g. non-TOP mTOR-sensitive, p-eIF4E-sensitive) against all other expressed genes (background) using a Wilcoxan rank sum test. This allows identification of significant shifts in the distributions that have a clear directionality (if there is an overall increase, or decrease in fold changes of FPs or total mRNA compared to background). If log2 fold changes are different from background, but without a clear directionality (equally likely to be increased or decreased), the test will not yield a significant result. This approach allows assessment of the overall behavior of gene signatures within a given dataset in a manner that is completely threshold-independent, such that it does not rely on classification of genes into different regulatory categories (translation only, buffering, etc.) based on significance or fold-change cut-offs (as in S4H). Therefore, we believe that this unbiased approach is well-suited for identifying cases when there are many genes that follow similar patterns of regulation within a given dataset.

(5) Page 10-"These results suggest that eIF3h depletion impacts the translatome differentially than depletion of eIF3e or eIF3d" ...These results suggest that eIF3h has less impact on the translatome, not that it does so differently. If it were changing translation by a different mechanism, I would not expect it to cluster with control.

This sentence was rewritten as follows: “The PCA plot and hierarchical clustering (Figure 2A and Supplementary Figure 4A) showed clustering of the samples into two main groups: RiboSeq and RNA-seq, and also into two subgroups; NT and eIF3hKD samples clustered on one side and eIF3eKD and eIF3dKD samples on the other. These results suggest that the eIF3h depletion has a much milder impact on the translatome than depletion of eIF3e or eIF3d, which agrees with the growth phenotype and polysome profile analyses (Supplementary Figure 1A and 1D).”

Other minor issues:(1) There are some typos: Figure 2 leves, Figure 4 variou,

Corrected.

(2) Figure 3, font for genes on volcano plot too small

Yes, maybe, however the resolution of this image is high enough to enlarge a certain part of it at will. In our opinion, a larger font would take up too much space, which would reduce the informativeness of this graph.

(3) Figure S5, highlighting isn't defined.

The figure legend for S5A (now S6A) states: “Less significant terms ranking 11 and below are in grey. Terms specifically discussed in the main text are highlighted in green.” Perhaps it was overlooked by this reviewer.

(4) At several points the authors refer to "the MAPK signaling pathway", suggesting there is a single MAPK that is affected, e.g in the title, page 3, and other places when it seems they mean "MAPK signaling pathways" since several MAPK pathways appear to be affected.

We apologize for any terminological inaccuracies. There are indeed several MAPK pathways operating in cells. In our study, we focused mainly on the MAPK/ERK pathway. The confusion probably stems from the fact that the corresponding term in the KEGG pathway database is labeled "MAPK signaling pathway" and this term, although singular, includes all MAPK pathways. We have carefully reviewed the entire article and have corrected the term used accordingly to either: (1) MAPK pathways in general, (2) the MAPK/ERK pathway for this particular pathway, or (3) "MAPK signaling pathway", where the KEGG term is meant.

(5) Some eIF3 subunit RNAs have TOP motifs. One might expect 3e and 3h levels to change as a function of 3d knockdown due to TOP motifs but this is not observed. Can the authors speculate why the eIF3 subunit levels don't change but other TOP RNAs show TE changes? Is this true for other translation factors, or just for eIF3, or just for these subunits? Could the Western blot be out of linear range for the antibody or is there feedback affecting eIF3 levels differently than the other TOP RNAs, or a protein turnover mechanism to maintain eIF3 levels?

This is indeed a very interesting question. In addition to the mRNAs encoding ribosomal proteins, we examined all TOP mRNAs and added an additional sheet to the S2 supplemental spreadsheet with all TOP RNAs listed in (Philippe et al., 2020, PMID: 32094190). According to our Ribo-Seq data, we could expect to see increased protein levels of eIF3a and eIF3f in eIF3dKD and eIF3eKD, but this is not the case, as judged from extensive western blot analysis performed in (Wagner et. al 2016, PMID: 27924037). Indeed, we cannot rule out the involvement of a compensatory mechanism monitoring and maintaining the levels of eIF3 subunits at steady-state – increasing or decreasing them if necessary, which could depend on the TOP motif-mediated regulation. However, we think that in our KDs, all non-targeted subunits that lose their direct binding partner in eIF3 due to siRNA treatment become rapidly degraded. For example, co-downregulation of subunits d, k and l in eIF3eKD is very likely caused by protein degradation as a result of a loss of their direct binding partner – eIF3e. Since we showed that the yeast eIF3 complex assembles co-translationally (Wagner et. al 2020, PMID: 32589964), and there is no reason to think that mammalian eIF3 differs in this regard, our working hypothesis is that free subunits that are not promptly incorporated into the eIF3 complex are rapidly degraded, and the presence or absence of the TOP motif in the 5’ UTR of their mRNAs has no effect. As for the other TOP mRNAs, translation factors eEF1B2, eEF1D, eEF1G, eEF2 have significantly increased FPs in both eIF3dKD and eIF3eKD, but we did not check their protein levels by western blotting to conclude anything specific.